# Black Soldier Fly Larvae as a Novel Protein Feed Resource Promoting Circular Economy in Agriculture

**DOI:** 10.3390/insects16080830

**Published:** 2025-08-10

**Authors:** Hongren Su, Bin Zhang, Jingyi Shi, Shichun He, Sifan Dai, Zhiyong Zhao, Dongwang Wu, Jun Li

**Affiliations:** 1Yunnan Provincial Key Laboratory of Animal Nutrition and Feed, Faculty of Animal Science and Technology, Yunnan Agricultural University, Kunming 650201, China; shr2904317687@163.com (H.S.); shijingyi3@163.com (J.S.); heshichun0529@163.com (S.H.); 15987179618@163.com (S.D.); 2Yunnan Academy of Animal Husbandry and Veterinary Sciences, Kunming 650201, China; binzhang89@163.com (B.Z.); zhaozhiyong988@163.com (Z.Z.)

**Keywords:** black soldier fly larvae (BSFL), animal feed, sustainable agriculture, nutritional value, growth performance

## Abstract

Against the backdrop of growing global demand for animal feed, traditional protein feeds face issues of supply shortages and high environmental costs. *Hermetia illucens* larvae (BSFL) offer a highly promising solution: these tiny larvae have a protein content as high as 40–60% and can serve as nutritious feed for poultry, fish, livestock, and pets, supporting their healthy growth. Notably, they can convert organic wastes such as kitchen garbage and agricultural residues into valuable resources, perfectly aligning with the eco-friendly concept of circular economy. However, key challenges remain—ensuring no heavy metal accumulation to guarantee safe use, establishing breeding standards, and enhancing consumer acceptance. Overall, black soldier fly larvae hold significant potential in alleviating feed shortages and improving ecological health.

## 1. Introduction

With the escalating global population, feed resource scarcity has intensified, particularly for protein sources where traditional ingredients face biological and geopolitical constraints. As traditional protein sources approach ecological carrying capacity thresholds, the FAO’s projections of accelerating feed demand exposes critical vulnerabilities in monoculture-dependent production systems. The current crisis is further exacerbated by the metabolic constraints of livestock species when it comes to utilizing plant-derived proteins [1].

Against this backdrop, insect protein, with its exceptional nutritional value and resource conversion efficiency, has emerged as an innovative alternative to traditional feed proteins, attracting widespread global attention. Studies have shown that insects represented by mealworms (*Tenebrio molitor*) have a crude protein (CP) content of up to 61% and possess a balanced amino acid profile and bioactive fats, which can precisely meet the nutritional requirements of livestock, poultry, aquatic, and pet feeds [2]. Among them, Black Soldier Fly Larvae (*Hermetia illucens* larvae, BSFL) further demonstrate “dual-cycle” industrial value: their breeding cost is 30% lower than that of traditional proteins, with a cycle of only 15–18 days per generation and a 65% reduction in the carbon footprint, combining economic and environmental advantages [3]. Meanwhile, they can efficiently process organic wastes such as food waste and agricultural by-products at a conversion rate of 1:10, simultaneously producing high-value-added animal protein and organic fertilizer [4]. This closed-loop model of “waste–feed–agricultural products” not only promotes insect ingredient suppliers to become core providers of sustainable feed but also facilitates the coordinated development of animal husbandry and environmental protection industries [5]. It is worth noting that BSFL can improve the feed conversion ratio of animals by 20%, and the antimicrobial peptides (AMPs) and lauric acid they contain can enhance animal immunity without affecting meat quality, achieving a win–win situation of economic and ecological benefits [6,7]. Li et al. [8] studies have confirmed that BSFL excel in improving animal growth performance, feed efficiency, and immune function. In addition to high nutritional value, their environmental advantages are also prominent: they can efficiently treat organic wastes such as animal manure and food waste, significantly reducing environmental pollution and greenhouse gas emissions [9,10].This dual benefit of resource conversion enables BSFL to occupy a key position in modern agriculture and the circular economy.

Despite the recognized benefits of insect protein, its widespread adoption still faces multiple obstacles. A primary issue is the significant gap in comprehensive regulatory frameworks that ensure the safety and quality of insect-derived foods, which must systematically address potential risks such as contamination by pathogens, heavy metals, and allergens. Secondly, consumer acceptance, particularly in Western societies, where entomophagy is non-traditional, remains a major challenge. This underscores the necessity for innovative processing technologies and effective marketing strategies to enhance product palatability and appeal [11].

To mitigate food safety risks, the substrates used for rearing BSFL require special attention, with animal-derived substrates and feces posing particularly high hazards. For example, EU Regulation (EC) No. 767/2009 explicitly prohibits the use of certain organic wastes (such as fish offal, feces, and municipal solid waste) in feed to ensure feed safety and reduce potential risks, aligning with the EU’s stringent food chain safety protocols [12].

Beyond traditional fields such as livestock and aquatic farming, the versatility of BSFL is driving transformative innovations in ruminant nutrition and pet food science. In ruminant production, the ingredients derived from BSFL are not only sustainable protein substitutes but also can regulate the dynamics of the rumen microbiome, optimizing the volatile fatty acid profile to improve energy utilization efficiency [13]. In the pet food sector, their balanced essential amino acid composition and bioactive lipids (including oleic acid and linoleic acid) can precisely meet the special needs of companion animals, especially in formulas related to immune support and skin health [14]. These cross-domain applications highlight their triple value of “excellent nutrition + environmental sustainability + economic feasibility”, making them a core cornerstone of future feed ingredient innovation.

This review systematically analyzes the nutritional composition of BSFL, examining the synergistic role of high-quality proteins and bioactive lipids in promoting growth, immune resilience, and metabolic health across various livestock species. It explores the key functions of inherent AMPs and gut microbiota-regulating compounds in reducing antibiotic dependence in animal feed—aligning with global trends toward sustainable and resilient production systems. By integrating mechanistic insights from rumen fermentation studies, pet nutrition trials, and diverse livestock applications (including poultry, swine, and aquaculture), this study provides a comprehensive framework for BSFL integration into modern feed systems.

## 2. Waste Nutritional Profile and Comparative Advantages of BSFL

### 2.1. BSFL-Derived High-Value Feed: From Organic Substrates to Sustainable Nutrition

BSFL have become an ideal alternative for the protein component in livestock feed due to their rich nutritional content and efficient organic matter conversion ability. BSFL digestive systems contain highly active enzymes (e.g., amylase, lipase, protease), which efficiently convert organic waste into high-value protein biomass [15]. Studies show that CP content in the dry weight of BSFL can reach up to 50%, with lipid content as high as 35%, and its amino acid profile is similar to that of traditional high-quality protein sources like fish meal (FM) [16]. Based on dry matter (DM) composition [17], the main components of BSFL include ether extract (EE) (15.0–34.8%), crude fiber (CF) (7.0–10%), ash (14.6–28.4%), and gross energy reaching 5278.49 kcal/kg, indicating its high nutritional value as a feed protein source [3].

BSFL protein consists predominantly of true protein (amino acids), with minor non-protein nitrogen components, ensuring a highly bioavailable and balanced amino acid profile for animal feed [18]. Additionally, BSFL’s lipid composition offers significant advantages. While insects generally have low Omega-3 fatty acid content, BSFL can accumulate these bioactive compounds—critical for aquatic animal growth and health—when reared on specific substrates (e.g., flaxseed oil or brown algae) [19]. Research shows that feeding larvae with fish viscera-enriched substrates elevates Omega-3 levels in both larvae and pre-pupae stages [20]. Notably, excessive inclusion of insect meal with suboptimal lipid profiles may reduce Omega-3 levels in farmed fish by up to 50%, underscoring the importance of targeted substrate management for maintaining nutritional quality [21]. In the protein of the pre-pupae, in BSFL, lysine, valine, and arginine are among the most abundant essential amino acids, with contents ranging from 20 to 30 g/kg DM, as reported in studies analyzing their nutritional profile. Among the non-essential amino acids, aspartic acid and glutamic acid are the predominant components [22].

BSFL can easily grow and spread on any nutrient substrate, such as vegetation residues, feces, animal manure, food scraps, agricultural by-products, or straw, whose composition is influenced by the growth substrate [22]. As shown in Table 1, BSFL were grown on two different substrates, such as kitchen waste (KM) and chicken manure (CM), with the aim of assessing their impact on nutrient composition. Research indicates that the protein content of larvae cultured on CM substrate was significantly higher (41.1% > 33.0%), while KM substrate promoted fat accumulation (34.3% > 30.1%). In terms of amino acid composition, the KM group had a richer content of phenylalanine (4.6% > 1.9%) and methionine (7.9% > 6.1%) [23]. Regarding minerals, the CM group had better calcium content (3.2% > 2.0%) and calcium–phosphorus ratio (5.2 > 8.3), while the KM group had higher potassium (5.7% > 4.9%) and iron (2.2% > 0.6%) contents. Fatty acid analysis showed that the CM group had higher arachidonic acid (6.7% > 5.7%) content, while the KM group had richer linoleic acid (7.5% > 5.8%) [17]. These differences are mainly due to the bioavailability of nutrients in the substrate and the metabolic selectivity of larvae, providing a theoretical basis for targeted cultivation.

It is worth noting that when BSFL uses solid aquaculture waste as the sole source of feed, heavy metals such as cadmium, mercury, manganese, and silver can be detected in their bodies and excreta. Among them, the accumulation levels of cadmium (bioaccumulation factor BSFL 2.5–2.7) and mercury (BSFL 1.6–1.9) are particularly significant [23]. Although some studies have not found a general trend of high accumulation, under specific conditions, certain elements (such as cadmium, with a BSFL of up to 7.1) may still reach potential risk levels. In addition, larvae exhibit significant physiological accumulation characteristics for various trace elements (such as barium, copper, iron, mercury, zinc, etc.), while elements such as aluminum, arsenic, and lead have not been observed to accumulate significantly [24].

Although BSFL exhibit certain physiological tolerance to heavy metals such as cadmium and zinc (e.g., body weight is not significantly affected), high concentrations of cadmium can still lead to delayed development or increased mortality rates. Therefore, in practical applications, it is necessary to seek a balance between resource conversion efficiency and safety risks, and to prioritize the use of raw materials with controllable heavy metal contamination as feed substrates [25]. In summary, although the heavy metal accumulation ability of BSFL can help in the biological detoxification of organic waste, the safety of using them as feed raw materials still needs to be carefully assessed. It is recommended to combine feed safety regulatory standards and take measures such as optimizing raw material selection, controlling feeding conditions, and post-treatment to effectively avoid potential risks.

**Table 1 insects-16-00830-t001:** Nutritional components of BSFL in different treatment matrices (%).

Processing Matrix	KM	CM
Moisture	78.41 ± 0.42	\
CP	33.0 ^b^ ± 1.0	41.1 ^a^ ± 0.3
EE	34.3 ^b^ ± 0.4	30.1 ^a^ ± 0.4
Ash	9.6 ± 1.6	9.3 ± 1.8
OM	\	59.8 ^a^ ± 0.4
Ca	2.0 ^b^ ± 1.41	3.2 ^a^ ± 2.32
P	4.1 ± 0.33	3.9 ± 0.31
Ca:P	8.3	5.2
k	5.7 ^b^ ± 0.04	4.9 ^a^* ± 0.08
Ma	3.3 ^b^ ± 0.06	4.0 ^a^ ± 0.34
Na	2.0 ± 0.09	2.4 ± 0.12
Fe	2.2 ^b^ ± 0.00	0.6 ^a^ ± 0.43
Cu	0.2 ^a^ ± 0.00	0.4 ^a^ ± 0.00
Isoleucine	2.6 ± 4.5	1.6 ± 1.5
Leucine	2.9 ± 5.2	3.0 ± 5.2
phenylalanine	4.6 ^b^ ± 4.7	1.9 ^a^ ± 2.4
Lysine	4.7 ± 0.5	4.1 ± 0.6
Methionine	7.9 ± 0.8	6.1 ± 0.8
Lauric acid	7.1 ^a^* ± 1.0	7.4 ^a^ ± 9.0
Linoleic acid	7.5 ± 0.1	5.8 ± 0.3
Linolenic acid	5.5 ^ab^ ± 0.5	5.6 ^a^ ± 0.1
Arachidonic acid	5.7 ^b^ ± 0.1	6.7 ^a^ ± 0.3
References	[16,26]	[16]

Note: \ represents that the original text is not given. CP: crude protein; EE: ether extract; Ash: crude ash; CM: chicken manure; KM: kitchen waste mixture; OM: organic matter. Different letter superscripts (^a,b^) indicate significant differences between groups (*p* < 0.05). The asterisk (*) is commonly used to indicate a statistically significant difference. A single asterisk (*) generally indicates statistical significance at the level of 0.01 < *p* ≤ 0.05.

### 2.2. Comparative Analysis of Nutritional Characteristics and Composition in BSFL Under Different Substrate Conditions

As presented in Table 2, the CP content of BSFL reared on KM was 23.24%, while that of BSFL reared on CM was slightly higher (25.2%). However, both were lower than soybean meal (SM, 46.8%) and FM (53.5%) [27,28,29]. On the other hand, the EE content of BSFL (9.19–12.77%) was significantly higher than that of SM (1.0%) and comparable to FM (10.0%), making it a potential high-energy feed source. Additionally, the CF content of BSFL (5.7–22.18%) was generally higher than that of FM (0.8%) and SM (3.9%), suggesting that BSFL may be more suitable for animal feed formulations requiring moderate fiber content. Notably, the ash content of BSFL reared on chicken manure was extremely high (43.33%), which may limit its application, whereas the ash content of BSFL reared on KM (4.33%) was closer to that of SM (4.8%), making it more practical for use [29,30].

### 2.3. Nutritional Composition Comparison Between BSFL and Other Insects

The nutritional composition of BSFL demonstrates remarkable stage-specific plasticity, establishing them as a versatile sustainable protein resource. As can be seen from Table 3, full-fat black soldier fly larvae (FF BSFL) exhibit a unique macronutrient profile, containing 43.1% CP and a notably high 38.6% EE, with saturated fatty acid (SFA) accounting for 70.72%. This high energy density provides significant advantages in monogastric animal feeds [23,31]. After drying, the nutritional components of degreased black soldier fly larvae (DF BSFL) undergo significant changes: the CP content increases to 51.83%, while EE decreases by 62% to 14.71%, reflecting lipid loss during processing, and the proportion of SFA remains high at 65.01% [31,32]. This stage-dependent nutrient redistribution distinguishes BSFL from other edible insects—such as yellow mealworms (*Tenebrio molitor*) (58.09% CP) and grasshoppers (*Acrida cinerea*) (78.7 g/kg leucine)—enabling the adjustment of protein–lipid ratios through simple processing methods [33]. Notably, BSFL contain 6.7% chitin, a key anti-nutritional factor, which is lower than the 8.91% in mealworms but still requires demineralization to improve digestibility [34]. Although their amino acid profile is less remarkable in essential amino acids (e.g., 3.30 g/kg lysine in DF BSFL), the high energy density from lipids compensates for this deficiency, making them ideal for formulating concentrated feeds. Compared with fly maggots (*Musca domestica*) (50% CP, 2.7% EE) and silkworms (*Bombyx mori*) (54% CP) [29,34], BSFL’s dual-stage nutritional adaptability—fresh for lipid-rich applications and dried for protein enhancement—exhibits unique advantages in circular economy models. Their ability to convert organic waste into high-energy biomass further enhances their sustainability. Future research should prioritize optimizing drying processes to preserve amino acid integrity and developing chitin modification technologies to unleash their full feed potential.

## 3. Intestinal Microbiota and Its Metabolic Pathways in BSFL

The unique nutritional benefits of BSFL are closely associated with their gut microbiota and metabolic processes. Research indicates that the BSFL intestinal tract harbors a highly diverse microbial community. These microorganisms serve as an ecological link between the host and the external environment, converting complex macromolecules from the surroundings into small-molecule nutrients directly utilizable by the host through intricate metabolic networks. This process plays an indispensable role in the host’s nutrient acquisition [33]. A thorough understanding of these intrinsic mechanisms is crucial for fully appreciating the value of BSFL in livestock applications.

Studies have shown that the gut microbiota structure of BSFL is regulated by multiple environmental and biological factors, including temperature, humidity, feed composition, and larval developmental stage [37]. Among them, feed ingredients as key environmental drivers have a significant context-dependent impact on the dominant microbial community. Comparative studies have shown that when rabbit manure is used as a substrate, the *Proteobacteria* become the dominant community. Although there has been observed excessive proliferation of *Campylobacter*, it has not had a negative impact on larval growth [38]. Under other substrate conditions such as poultry manure or pig manure, the dominant bacterial groups may be *Firmicutes* or *Bacteroidetes* [39]. The variability in response to this microorganism highlights the complexity of feed composition regulation. Temperature factors also play a significant role in regulating microbial community structure, with studies finding that the abundance of the genus *Providencia* decreases with increasing temperature, while the genera *Alcaligenes*, *Bacteroides*, and *Enterobacter* show a significant increase trend [40]. From the perspective of developmental dynamics, systematic analysis based on 16S rRNA gene sequencing indicates that the microbial community structure shows characteristic changes at different developmental stages such as eggs, larvae, pupae, and adults. Among them, the *Proteobacteria* phylum occupies a dominant position throughout the entire life cycle (especially during the egg stage), while the diversity of microbial communities during the larval and pupal stages is significantly higher than that during the egg and adult stages [41]. Ao et al. [39] conducted a systematic study on the dynamic evolution of gut microbiota during the treatment of CM and pig manure by BSFL using 16S rRNA sequencing technology. They found that the *Proteobacteria*, *Firmicutes*, and *Bacteroidetes* phyla were dominant, with the *Bacteroidetes* significantly increasing during the manure treatment process. Further analysis identified 10 core genera significantly related to key nutrients such as total nitrogen and crude fiber in the manure (such as *Prevotella* and *Enterococcus*), and functional prediction showed that these microorganisms highly expressed genes related to carbohydrate and amino acid metabolism, suggesting they may play an important role in promoting the efficient conversion and accumulation of proteins by BSFL. These multi-substrate comparative study results emphasize that when explaining the variation patterns of BSFL gut microbiota, the impact of specific feed systems must be fully considered to avoid overgeneralization of conclusions.

The BSFL gut microbiota not only plays a key role in maintaining host physiological functions, immune regulation, and nutritional metabolism but also is deeply involved in core metabolic processes such as protein degradation, energy metabolism, and amino acid synthesis (as shown in Figure 1). The core microbiota, represented by *Enterococcus*, *Klebsiella*, *Morganella*, *Providencia*, and *Lactobacillus*, can significantly enhance protein degradation efficiency by secreting highly effective proteolytic enzymes [42].

Notably, BSFL gut microorganisms not only degrade proteins and convert substrates into metabolizable energy but also synthesize amino acids. This functional capability of the resultant microbial protein is commonly observed in gut symbionts of insects and other animals [36]. With the continuous innovation of technical means, researchers have deeply revealed the specific molecular mechanisms of the BSFL gut microbiota in metabolic activities such as protein degradation and amino acid synthesis [43]. The study by Luo, X. et al. [44] for the first time confirmed that the non-pathogenic gut symbiont *Citrobacter amalonaticus* can significantly promote the growth of BSFL. Its mechanism primarily involves regulating the Hitryp serine protease and Himtp metallopeptidase gene expression, thus boosting protein metabolism efficiency. This study innovatively developed a symbiont-mediated RNAi technology, successfully achieving in situ gene knockout and confirming that the above-mentioned genes are key targets for the growth-promoting effect of the microbiota. This research not only provides a new tool for studying the interaction mechanism between insects and microorganisms but also lays an important theoretical foundation for the development and application of biotransformation technologies.

Based on the above research results, by optimizing the feed matrix formula and precisely regulating the structure of the gut microbiota, BSFL have the potential to become an efficient and sustainable raw material for animal feed. The unique mechanisms of their gut microbiota in enhancing protein digestion and absorption efficiency offer innovative solutions and new research ideas for addressing global feed protein shortages and ensuring food security. Future research can focus on the functions of specific microbial communities where optimized and microbial community regulation techniques were developed to further explore the potential application of BSFL in agricultural ecosystems.

## 4. Applications of BSFL in Animal Production

With the exponential growth of animal husbandry (poultry, swine, aquaculture), protein feed demand has surged, while traditional sources (SM, FM, blood plasma protein) face critical challenges: import-dependent costs, supply volatility, and biosecurity risks. BSFL offer a paradigm shift, featuring high protein content, comparable to conventional proteins; cost-effective production via organic waste utilization; and a sustainable lifecycle integrating waste remediation with protein synthesis [45]. Research shows BSFL diets improve growth, feed efficiency, and immunity across species, positioning them as a transformative solution for balancing nutritional efficacy and environmental sustainability in modern animal agriculture (as shown in Figure 2).

### 4.1. Application in Poultry Production

BSFL as a novel sustainable protein source demonstrate broad application prospects in poultry farming. Figure 3 illustrates the relationship between BSFL and broiler weight gain. As shown in Table 4, the optimal inclusion level of BSFL exhibits species specificity: 20% in broiler feed, 5–25% in laying hens, 20% in quails, 9% in Muscovy ducks, 1% in geese, and no more than 5% in turkeys. Additionally, processing methods (full-fat/defatted) and feeding regimes (single/mixed feeding) significantly influence its efficacy. Appropriate inclusion of BSFL can markedly enhance growth performance, improve gut health status, and optimize the quality of meat and egg products. However, exceeding the safe threshold may lead to adverse effects such as metabolic disorders, intestinal morphological alterations, and reduced production performance. Future research should focus on optimizing precise inclusion ratios for different growth stages, exploring synergistic mechanisms with other feed ingredients, and developing nutritional balancing strategies for large-scale applications to ensure the safe and efficient utilization of BSFL in animal feed.

**Table 4 insects-16-00830-t004:** The impact of adding BSFL on poultry.

Poultry Species	Scientific Name	BSFL (%)	Impact	References
Broiler	\	4–25	Improve growth performance, feeding rate, and nutrient conversion efficiency; optimize gut microbiota and immune function without affecting blood parameters or health.	[46,47,48,49,50]
		30	Decreased intake of total DM and metabolizable energy, leading to a decline in protein utilization efficiency. Increased plasma uric acid and serum alkaline phosphatase concentrations.	[46]
		50–100	Reduced growth performance, chicken flavor, and breast muscle ratio when replacing soybean meal.	[51,52]
Laying hens	\	5–25	BSFL can effectively regulate the intestinal flora of laying hens, promote the production of short-chain fatty acids, improve gut health, and enhance the egg production rate and egg quality. At the same time, it does not affect egg weight, shell quality, or feed efficiency and has no negative impact on liver metabolism and overall health.	[53,54,55,56]
		50	Decreased digestibility of dry matter, organic matter, and crude protein (possibly due to negative effects of chitin). Reduced serum cholesterol and triglycerides, increased serum globulins, and decreased albumin/globulin ratio.	[53]
Quail	*Coturnix japonica*	20	Significantly improve quail production performance, enhance liver and kidney function and metabolism–antioxidation–immune regulation, optimize egg quality, and promote the digestion and absorption of proteins and minerals.	[57,58]
Mandarin duck	*Cairina moschata domestica*	9	Defatted BSFL have no adverse effect on the slaughter characteristics and meat quality of Muscovy ducks. However, they changed the fatty acid profile.	[59]
Sichuan White Goose	\	1	Significantly improve the daily weight gain of geese, enhance immunity (increase antibody titer and IgG/C3 levels, promote IL-6 and CD4 expression), and improve intestinal function (enhance barrier, regulate microbial balance).	[60]
Turkey	*Meleagris gallopavo*	5	No adverse effects on liver lipid status and histology.	[61]
		10–15	Disturbs lipid metabolism and increases cholesterol, lipid oxidation, and liver fat deposition	

**Figure 3 insects-16-00830-f003:**
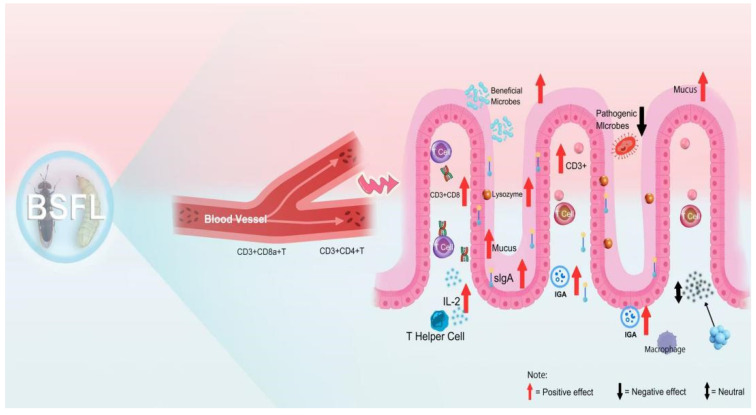
Influence of BSFL as a dietary supplement on the immune response and performance of broiler chickens [62].

### 4.2. Application in Pig Production

Numerous studies have confirmed that BSFL, as a functional protein source, hold significant application value in pig farming. Yu et al. [63] showed that replacing 50% of FM with BSFL meal could significantly improve the intestinal health of weaned piglets. It increased the abundance of probiotics such as *Lactobacillus* and *Bifidobacterium* in the ileum and cecum; elevated the contents of lactate and short-chain fatty acids; and simultaneously upregulated the expression of genes related to intestinal mucosal barrier function and development, enhanced anti-inflammatory factors, and inhibited inflammatory pathways, thereby comprehensively promoting intestinal health. Further studies found that adding 10% BSFL to the diet of piglets had a positive impact on the cecal microbiota and small intestinal mucus dynamics [64].The probiotic effects of BSFL mainly stem from their rich AMPs. Studies have shown that BSFL-derived AMPs possess the following prominent characteristics: (1) small molecular weight, facilitating absorption and function; (2) excellent thermal stability (able to withstand high-temperature treatment at 100 °C); (3) broad-spectrum antibacterial activity, especially showing significant inhibitory effects on clinically common drug-resistant strains (such as *MRSA*, multidrug-resistant *Escherichia coli*, etc.); (4) a unique mechanism of action, exerting bactericidal effects by destroying the integrity of bacterial cell membranes, which is not easy, to induce bacterial resistance. These characteristics make BSFL-AMPs a new type of antibacterial substance with great development potential [7]. Adding 12% BSFL to the diet can not only improve the growth performance (daily weight gain increased by 18.7%) and nutrient utilization rate of piglets but also inhibit the expression of virulence genes of pathogenic bacteria (such as *Escherichia coli*) through AMPs, reduce the intestinal inflammatory response (TNF-α decreased by 32.4%), and enhance the antioxidant capacity (SOD activity increased by 25.6%) [65]. In addition, replacing FM and SM with BSFL can improve the apparent digestibility of nutrients in weaned piglets (CP digestibility increased by 6.8%), which is related to their characteristics of being rich in active peptides and chitin [66]. Driemeyer et al. [67] added 3.5% BSFL to the piglet diet without affecting the average daily gain, DM intake, and digestibility, but reduced the feed conversion rate.

Maureret al. [68] confirmed that partial replacement of SM with 8% BSFL had no negative impact on the growth performance of nursery pigs. However, it should be noted that compared with high-digestibility animal protein diets, its growth-promoting effect is not evident and even shows a slight negative impact (daily weight gain decreased by 7.2%). The latest research shows that adding 4% BSFL powder to the feed can significantly increase the average daily gain of growing pigs and improve backfat thickness. In terms of meat quality, both the 2% and 4% addition groups can significantly increase intramuscular fat content. Metabolomics analysis shows that it can increase free amino acid levels and significantly regulate lipid metabolism [69]. Yu et al. [70] found that adding 4% BSFL to the diet can not only improve the growth performance and carcass traits of finishing pigs but also significantly increase intramuscular fat deposition by upregulating the expression of fat synthesis genes.

To sum up, BSFL, as a new type of functional protein source, have shown multiple benefits in pig breeding: they can significantly improve the intestinal health of piglets by regulating intestinal flora, enhancing barrier function and anti-inflammatory activity; the contained AMPs have a unique antibacterial mechanism; adding 2–4% can optimize the production performance of growing and finishing pigs and improve meat quality. Studies have confirmed that BSFL can safely replace 50% FM or 8% SM, with both health-promoting and quality-improving effects, providing a sustainable protein solution for green pig breeding.

### 4.3. Application in Aquaculture

As a green alternative to FM and SM, BSFL have established their strategic position in the aquaculture feed protein system by optimizing animal health indicators and reducing the environmental footprint [71]. Studies indicate that incorporating BSFL into farmed animal (including aquatic species) diets at varying inclusion levels can modulate growth performance, immune function, and gastrointestinal health, primarily through gut microbiota regulation and bioactive compound interactions (Table 5).

A critical challenge in aquaculture is *Aeromonas hydrophila*, a pathogenic bacterium that causes enteritis and septicemia in fish and impairs shrimp vitality and feeding efficiency. To mitigate such infections, functional feed additives—particularly those with immunomodulatory and antimicrobial properties—have gained attention. Among these, BSFL have shown promise in enhancing disease resistance. For instance, supplementing golden pompano (*Trachinotus blochii*) diets with 1–3% BSFL improved innate immunity and growth performance, likely through BSFL-derived AMPs that selectively inhibit pathogens while fostering beneficial gut microbiota [72]. However, the broader applicability of these findings requires further validation, as BSFL’s effects may vary across aquatic species due to differences in the digestive physiology and gut microbiome composition. Additionally, while BSFL offer nutritional benefits, they contain approximately 9% chitin, an anti-nutritional factor known to impair growth and nutrient utilization when dietary levels exceed 1% [73]. This suggests that the net benefits of BSFL depend on balancing their immunostimulatory components (e.g., AMPs) against potential anti-nutritional effects. Future research should clarify this trade-off by investigating species-specific responses, optimal inclusion levels, and the mechanisms underlying BSFL’s dual roles in disease resistance and growth modulation. Such insights will advance BSFL as a sustainable aquaculture solution while minimizing unintended trade-offs [74].

**Table 5 insects-16-00830-t005:** The effects of BSFL on the growth performance, physiological functions, and health indicators of aquatic animals in daily feed.

Test Animals	Scientific Name	BSFL (%)	Main Function	References
Atlantic salmon	*Salmo salar*	6.25–12.5	Had a lower immune response to the skin mucus protein profile of Atlantic salmon. Showed good intestinal health and normal metabolic response.	[75,76]
Largemouth bass	*Micropterus salmoides*	1.0	Improve growth performance and disease resistance, enhance antioxidant capacity, and promote intestinal health and microbiota.	[77]
Spotted catfish	*Ictalurus punctatus*	\	Altering the overall gene expression and activating both innate and adaptive immunity may support the species’ disease resistance.	[78]
Red hybrid tilapia	*Oreochromis*	35–42	Growth performance and increase the content of crude protein and fat in vivo.	[79]
Rainbow trout	*Oncorhynchus mykiss*	4–10	Improved the weight, feed intake, and growth performance and immune response of rainbow trout.	[80]
Juvenile Pacific white shrimp	*Litopenaeus vannamei*	4.5–10.5	The body weight gain, feed conversion ratio, and specific growth rate of Pacific white shrimp showed a linear improvement with increasing inclusion levels.	[81]
Juvenile mud crab	*Scylla paramamosain*	25–50	BSFO can safely replace fish oil in partial amounts, not only maintaining normal growth performance in mud crabs but also enhancing antioxidant capacity, improving immune response, optimizing lipid metabolism, and promoting mitochondrial function, making it a promising sustainable alternative feed ingredient.	[82]
Asian swamp eel	*Monopterus albus*	24.1–32.8	Damage the digestion and absorption of nutrients in Asian swamp eels. Excessive BSFL may induce liver lipid accumulation and affect intestinal morphology.	[83]
Hybrid grouper	*Epinephelus fuscoguttatus* ♀ × *E. lanceolatus* ♂	30–50	The replacement of fish meal with BSFL will weaken the intestinal wall, leading to vacuoles, a sparse striated border, and reduced villi.	[84]

### 4.4. Application in Ruminants and Pets

BSFL are recognized for their high protein content and beneficial fatty acids, demonstrating significant potential as a novel functional protein feed in ruminant production. While BSFL have been validated as a feed alternative in poultry, pigs, and aquaculture, their application in ruminant nutrition remains under-researched, especially regarding their long-term impact on the rumen microbiota and the improvement mechanism of meat and milk quality. This research gap seriously restricts the promotion and application of BSFL in ruminant production.

While BSFL exhibit potential nutritional benefits, their application in ruminant diets requires careful consideration of chitin digestibility and fat composition, which may constrain inclusion levels. Multiple studies have demonstrated that the inclusion ratio of BSFL has a significant regulatory effect on rumen fermentation parameters. Research by Lu et al. [85] showed that adding 10–15% BSFL to ruminant diets can optimize rumen microbial activity by providing readily fermentable protein and fat, significantly increasing gas production and volatile fatty acid (VFA) yield while reducing methane emissions. The study suggested that precise control of the inclusion ratio (recommended at around 10%) is key to balancing nutritional enhancement and emission reduction.

In the application of BSFL oil (BSFO), a 4% inclusion rate combined with a 40:60 concentrate-to-forage ratio significantly reduced methane emissions, increased propionate production, and maintained DM degradability, whereas a 6% inclusion rate led to reduced digestibility due to fat overload [86]. This indicates that the BSFO inclusion ratio should be dynamically optimized based on production stages to achieve the best feeding efficiency.

Based on existing evidence, it is strongly believed that adding chitin and chitosan to the diets of ruminants (including dairy cows, beef cattle, goats, and sheep) has beneficial effects on meat and milk production, feed intake and digestibility, rumen fermentation, rumen pH, bacterial diversity, growth rates, and wool yield. From a nutritional perspective, their effects may be significant, particularly their ability to prevent rumen degradation, produce less ammonia nitrogen, and more effectively bypass proteins to the lower intestine. These qualities improve rumen fermentation by generating more propionate and less methane. Most importantly, they enhance ruminant productivity, namely, the yield of meat, milk, and wool. Additionally, chitin and chitosan exhibit immunomodulatory and antimicrobial properties when used as feed additives [87]. Nekrasov et al. [77] further found that adding 5% BSFL oil to dairy cow feed significantly increased the conjugated linoleic acid content in milk fat (15%), a mechanism potentially related to the upregulation of fatty acid transporter expression in rumen epithelial cells. However, current research has primarily focused on short-term effects (≤8 weeks), lacking systematic evaluation of long-term changes in rumen microbiota (e.g., methanogen dynamics) and heavy metal residues in milk. Additionally, the interaction mechanisms with the host metabolism need to be elucidated using metagenomic technologies.

Supplementation with BSFL can optimize the rumen fermentation environment, increase VFA production, and improve dairy cow performance. A low dose (10 g/day) enhances milk yield, while a high dose (100 g/day) more significantly increases the milk fat percentage. Therefore, BSFL fat serves as a functional fat source with potential application value in dairy cow nutrition regulation [88]. Braamhaar et al. [89] found that DF BSFL meal can feasibly replace SM in dairy cow diets. A 50% replacement ratio balances feed intake and digestion efficiency without negatively affecting milk production performance, providing a basis for reducing reliance on SM and establishing a circular feed system.

Astuti et al. [10] conducted a comparative study on the effects of goat milk, commercial milk replacer, and a milk replacer containing 30% 6-days-old BSFL meal on pre-weaned goat kids, which revealed that the BSFL-based replacer provided comparable DM and protein intake to goat milk, higher fat intake, and a superior feed conversion rate (69.16). Additionally, it had no adverse effects on kid physiology. As a protein supplement for beef cattle, the optimal inclusion level of BSFL was found to be 10–15% (on a DM basis), significantly improving the intake of low-quality roughage (CP < 7%)—with the forage OM intake reaching 4.30 kg/d—and the total digestible OM intake (3.24 kg/d), achieving a performance close to that of SM. BSFL supplementation did not negatively affect organic matter digestibility or rumen health (stable VFA and ammonia nitrogen levels). With its high protein (38% CP), high energy (109% TDN), and low-carbon production advantages (projected price: USD 312.30/short ton), BSFL serves as a sustainable alternative protein source. However, attention should be paid to its high fat content (35% DM) and its potential impact on rumen balance during the finishing phase [90].

Current research still faces three major limitations: (1) Short-term effect studies (≤8 weeks) dominate, with a lack of systematic evaluation of long-term rumen microbial community dynamics (e.g., methanogen shifts); (2) The risk of heavy metal residues in milk and the interaction mechanisms with the host metabolism remain unclear, requiring further elucidation through metagenomic technologies; (3) Before industrial-scale application, key optimizations in feed palatability, long-term feeding safety assessments, and economic feasibility evaluations are still needed. Breakthroughs in these areas will establish a more robust scientific foundation for the large-scale application of BSFL in ruminant farming.

BSFL have demonstrated unique advantages and challenges as a novel protein source in the pet nutrition field. From a nutritional perspective, although BSFL have a balanced amino acid composition, the chitin contained in their exoskeleton (with nitrogen content of 6.89% and a recovery rate exceeding 95% in protein matrices [91]) can lead to reduced apparent digestibility of CP. This characteristic suggests that in practical formulations, it may be necessary to supplement essential amino acids to ensure nutritional balance. In terms of safety, it is worth noting that insect proteins are not traditionally considered low-allergen ingredients; about 1% of dogs and cats may have an allergic reaction to them [92], especially due to cross-reactivity with crustacean allergens (including arginine, chitinase, tropomyosin, etc.). Studies have shown that mite-allergic dogs have a cross-reaction with yellow mealworm protein [93], indicating that even after heat treatment, caution must be maintained in its application in formulations.

Existing research has revealed differences in the application of different components of BSFL in pet food. Studies on fat components (Kahraman et al. [94]) have shown that replacing 6% poultry fat with BSFL fat can maintain normal serum biochemical indicators and immunoglobulin levels in experimental dogs, but it leads to a slight decrease in nutrient digestibility and palatability. Research on protein components, however, has shown significant dose–response differences: adding 10% FF BSFL can enhance palatability [95], while DF BSFL protein (replacing 8% CM) significantly reduces protein digestibility and causes dysbiosis in the hindgut microbiota, characterized by reduced short-chain fatty acids, increased pH values, and enrichment of potential pathogens such as *Clostridium difficile* and *Enterococcus*. Nevertheless, BSFL protein hydrolysates have shown potential to reduce IL-6 levels and improve intestinal inflammation by regulating the TLR4/NF-κB signaling pathway [96].

Research on pets in special physiological stages found that a 5% addition of BSFL had no adverse effects on the feeding behavior, fecal characteristics, and blood indicators of elderly dogs [97]. In particular, fermented BSFL combined with oats could further improve skin health scores [98]. A short-term (10 days) safety assessment showed that BSFL fat (replacing 5% mixed oil) performed well in maintaining serum parameters and the antioxidant capacity in experimental animals, while the defatted protein group exhibited disturbances in 18 fecal metabolic pathways [99]. It is worth noting that most current studies are limited to short-term trials, and regarding the long-term effects of consumption on pet kidney function and trace element balance, more systematic multi-generation toxicity tests are still needed.

## 5. Conclusions

BSFL have revolutionized animal production paradigms through their bioconversion plasticity and symbiotic relationships with gut microbiota. Their chitin–protein matrix can mediate cross-species metabolic regulation, establishing a synergistic system of “waste valorization–precision nutrition–environmental mitigation”. Relying on developmental stage-specific nutrient partitioning and modular microbial functionality, BSFL act as “living bioreactors” that optimize microbiota–immune interactions across farming systems while enabling ecological waste treatment. This study further confirms that BSFL contain 40–60% protein, with their nutritional composition precisely regulable through rearing substrates. They can efficiently convert organic waste into high-value biomass, showing remarkable performance in replacing traditional protein sources such as FM and SM. When appropriately incorporated into the feed of livestock and aquatic organisms, BSFL not only enhance animal growth performance and improve intestinal health and immunity but also reduce methane emissions and lower environmental footprints. The research innovatively reveals the mechanism by which their gut microbiota promotes protein accumulation through carbohydrate and amino acid metabolism, and identifies the potential of antimicrobial peptides and lauric acid in replacing antibiotics, providing new ideas for green farming. In practice, the closed-loop model of “waste–feed–agricultural products” constructed by BSFL offers a sustainable solution to the circular economy development and global protein shortage, driving the green transformation of agriculture. Despite challenges such as microbiota engineering, synthetic biology and ecological engineering can advance BSFL towards designed biological systems. Future research should focus on precise microbiota regulation to optimize metabolic efficiency; optimization of rearing substrates to control heavy metal accumulation and improve chitin digestibility; assessment of long-term feeding safety (especially regarding heavy metal residues and immune cross-reactions); and use of metagenomic technologies to analyze the interaction mechanisms between rumen microorganisms and hosts. By closing the “biotransformation–multitrophic utilization” loop, BSFL have emerged as a key to solving global protein and ecological crises, leading an entomological paradigm shift in sustainable agroecosystems.

## Figures and Tables

**Figure 1 insects-16-00830-f001:**
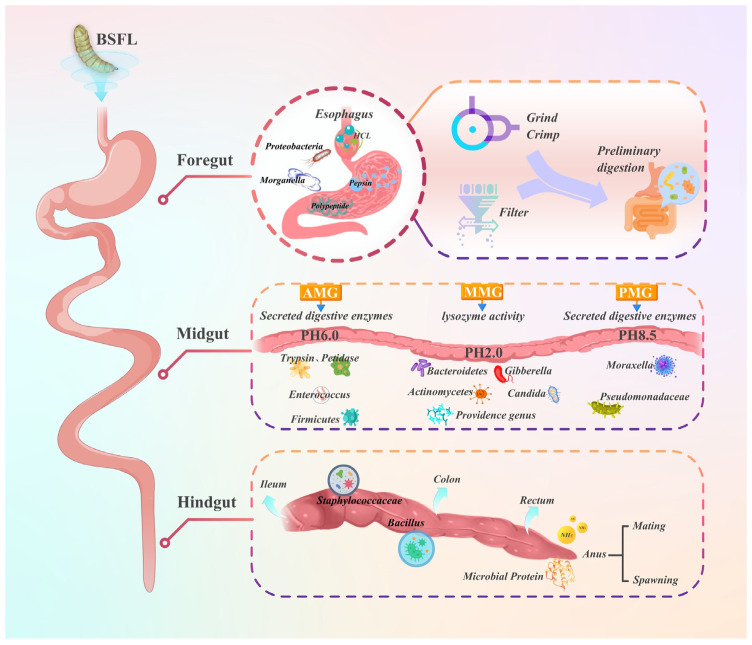
Illustrates the intestinal structure of BSFL and the distribution of their microbial communities.

**Figure 2 insects-16-00830-f002:**
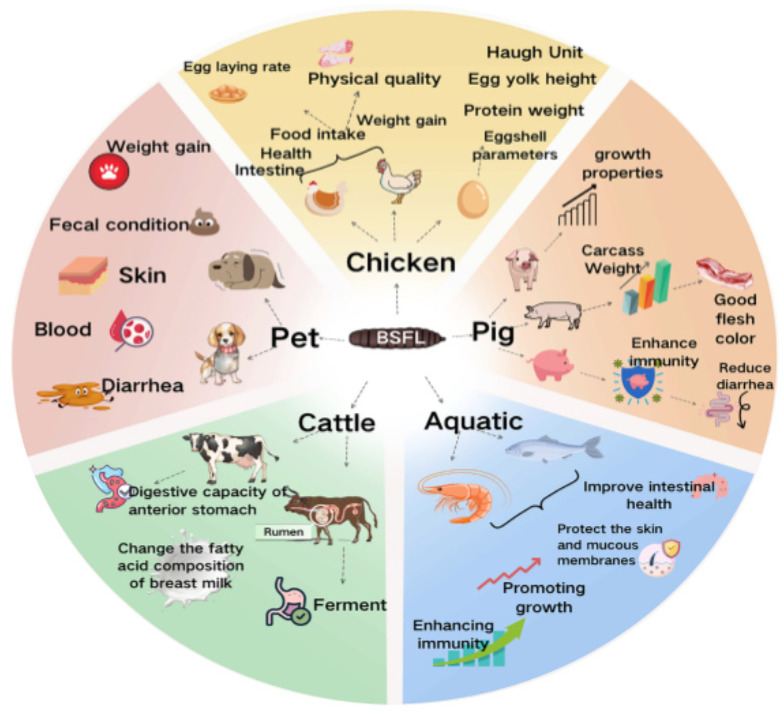
Application and effects of BSFL in animal production.

**Table 2 insects-16-00830-t002:** Comparison of BSFL with other protein sources (%).

Raw Material	Ash	CP	CF	EE	References
Kitchen waste BSFL	4.33	23.24	22.18	9.19	[26]
Chicken manure BSFL	43.33	25.2	5.7	12.77	[27]
SM	4.8	46.8	3.9	1.0	[28]
FM	20.8	53.5	0.8	10.0	

Note: The nutritional value of soybean meal and fish meal, including crude ash, crude protein, crude fiber, and crude fat, is sourced from the 3rd edition of the Chinese Feed Composition and Nutritional Value Table, as detailed in the referenced document. SM: soybean meal; FM: fish meal.

**Table 3 insects-16-00830-t003:** Comparison of BSFL nutrients with other insects.

Nutritional Composition	Insect Species					
	FF BSFL	DF BSFL	*Musca domestica*	*Tenebrio molitor*	*Acrida cinerea*	*Bombyx mori*
CP	43.10	51.83	50.0	53.0	58.09 ± 0.09	54.0
CF	\	\	18.9	3.1	4.46 ± 0.04	3.9
EE	38.6	14.71	2.7	3.6	8.88 ± 0.11	2.5
Ash	2.7	7.27	10.1	26.8	7.14 ± 0.16	5.8
SFA	70.72	65.01	\	20.99	25.0 ± 0.71	\
Isoleucine	1.91	2.24	3.2	4.6	43.7 ± 0.09	3.9
Threonine	1.62	1.93	3.78	4.0	\	4.8
Methionine	0.71	0.62	2.2	1.5	20.6 ± 0.06	3.0
Phenylalanine	1.64	1.69	4.6	4.0	32.6 ± 0.19	4.4
Lysine	2.3	1.96	6.1	5.4	15.5 ± 0.13	6.1
Chitin	6.7	\	\	8.91	\	\
References	[22,29]	[30,31]	[32]	[32,33,34,35]	[36]	[29]

Note: FF BSFL: full-fat black soldier fly larvae; DF BSFL: degreased black soldier fly; SFA: saturated fatty acid.

## Data Availability

No new data were created or analyzed in this study. Data sharing is not applicable to this article.

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
