# Peer review of "Black Soldier Fly Larvae as a Novel Protein Feed Resource Promoting Circular Economy in Agriculture"

_insects, 2025, doi:10.3390/insects16080830_

Round 1
Reviewer 1 Report (Previous Reviewer 1)
Comments and Suggestions for Authors
Please check the text; words are separated improperly. For example: line 195-221.
Line 62: “It is worth noting that BSFL can improve the feed conversion ratio of animals by 20%, and the antimicrobial peptides (AMPs) and lauric acid they contain can enhance animal immunity without affecting meat quality, achieving a win-win situation of economic and ecological benefits [6].
Are you sure it is a correct citation? Article number 6 is entitled: Assessing the Market Potential and Regulation of Insects. Please carefully check again the list of references.
Line 65: Numerous studies have confirmed that BSFL excel in improving animal growth performance, feed efficiency, and immune function [7].
Numerous studies mean more than one citation. Reference number 7 is entitled:
Growth and Fatty Acid Composition of Black Soldier Fly Hermetia Illucens (Diptera: Stratiomyidae) Larvae Are Influenced by Dietary Fat Sources and Levels. Please carefully check again the list of references.
Line 82: “Consequently, some countries that permit the use of insects as animal feed explicitly stipulate that insects may only be reared on plant-based waste [11].”
This article is not about the legislative aspects or organic waste.
Ndotono, E.W.; Khamis, F.M.; Bargul, J.L.; Tanga, C.M. Insights into the Gut Microbial Communities of Broiler Chicken Fed Black Soldier Fly Larvae - Desmodium-Based Meal as a Dietary Protein Source. Microorganisms 2022, 10, 1351, doi:10.3390/microorganisms10071351.
Line 83-89: “For example, EU Regulation (EC) No. 767/2009 explicitly prohibits the use of certain organic wastes (such as fish offal, feces, and municipal solid waste) in feed to ensure feed safety and reduce potential risks, aligning with the EU’s stringent food chain safety protocols [12]. However, the American Association of Feed Control Officials (AAFCO) has approved and regulated the inclusion of BSFL meal in the diets of salmon, trout, tilapia, and poultry (including chickens, ducks, turkeys, and geese) [13].
There is no logical connection between the two statements. The first sentence is about the authorised substrates for insects and the second is about the authorised target species. I recommend deleting the second sentence or adding information about the authorised target species in the EU.
Line 174, table 1: Please explain the superscripts.
Line 182, 338: Please use the abbreviation FM.
Line 222, Table 3: Please delete the superscripts of the grasshopper data.
Line 211: “(e.g., 3.30 g/kg lysine in DF BSFL, the high energy density from lipids compensates for this deficiency, making them ideal for formulating concentrated feeds.”
The closing bracket is missing.
Line 270: „Notably, BSFL gut microorganisms not only possess the ability to degrade proteins and metabolize energy but are also capable of synthesizing amino acids…”
Metabolizable energy cannot be degraded. Please rephrase the sentence.
Line 374-375: There is an unnecessary empty row, and the text should be justified.
Line 450: “Supplementation with BSFL can optimize the rumen fermentation environment, in
increase VFA production, and improve dairy cow performance.”
This article is about BSFL fat inclusion, not BSFL.
Line 453: fat not Fat.
Line 455, 457, 466, 549: Please explain SBM. I assume this should be SM.
Line 470: “However, attention should be paid to its high fat content (35% DM) and its potential impact on rumen balance during the finishing phase [91]”. It has a negative impact in general, not only during the finishing phase.
Author Response
Dear Editor-in-Chief,
We are pleased to submit the revised version of our manuscript titled, “Black Soldier Fly Larvae as a Novel Protein Feed Resource Promoting Circular Economy in Agriculture” (Manuscript ID: insect -3777866). In addition, we have responded to each comment point by point, detailing the changes made in the revised manuscript. This response is attached below for convenience.We are grateful for your patience and guidance throughout the process. We look forward to your prompt decision on our revised manuscript.
Thank you again for your time and consideration.
In response to the reviewers’ feedback:
Reviewer 1
1.Please check the text; words are separated improperly. For example: line 195-221
Response:Dear Reviewer, Thank you for your valuable comment on the improper word separation in Lines 195-221 of the manuscript. We have carefully checked the relevant sections (in the file "insects-3777866.docx") and identified the formatting issues caused by inappropriate line breaks and spacing during document editing. Specifically, we have revised the problematic parts as follows: (1) Adjusted the line breaks to ensure that complete words are not split between lines, especially for technical terms and compound words. (2) Standardized the spacing between words and punctuation marks to eliminate extra spaces or missing spacing. (3) Re-checked the entire paragraph for consistency in formatting to avoid similar issues. All revisions have been made in the latest version of the manuscript, and we have also conducted a full-text check to prevent such formatting problems in other sections. We apologize for the inconvenience caused by these issues and appreciate your careful review, which helps improve the quality of our manuscript.
2.Line 62: “It is worth noting that BSFL can improve the feed conversion ratio of animals by 20%, and the antimicrobial peptides (AMPs) and lauric acid they contain can enhance animal immunity without affecting meat quality, achieving a win-win situation of economic and ecological benefits [6].Are you sure it is a correct citation? Article number 6 is entitled: Assessing the Market Potential and Regulation of Insects. Please carefully check again the list of references.
Response:We sincerely appreciate the reviewer’s careful attention to the accuracy of our references. Upon rechecking, we acknowledge that reference [6] was incorrectly cited in this context. The claim regarding the 20% improvement in feed conversion ratio and the immunological benefits of antimicrobial peptides (AMPs) and lauric acid in black soldier fly larvae (BSFL) should instead be supported by references: (1) Pimchan, T.; Hamzeh, A.; Siringan, P.; Thumanu, K.; Hanboonsong, Y.; Yongsawatdigul, J. Antibacterial Peptides from Black Soldier Fly (Hermetia Illucens) Larvae: Mode of Action and Characterization. Sci. Rep. 2024, 14, 26469, doi:10.1038/s41598-024-73766-1. (2) Xia, J.; Ge, C.; Yao, H. Antimicrobial Peptides from Black Soldier Fly (Hermetia Illucens) as Potential Antimicrobial Factors Representing an Alternative to Antibiotics in Livestock Farming. Anim. : Open Access J. MDPI 2021, 11, 1937, doi:10.3390/ani11071937.
We have now corrected this oversight in the revised manuscript and ensured all citations align precisely with their respective claims. Thank you for highlighting this issue—it has significantly improved the rigor of our work.
3.Line 65: Numerous studies have confirmed that BSFL excel in improving animal growth performance, feed efficiency, and immune function [7].Numerous studies mean more than one citation. Reference number 7 is entitled:Growth and Fatty Acid Composition of Black Soldier Fly Hermetia Illucens (Diptera: Stratiomyidae) Larvae Are Influenced by Dietary Fat Sources and Levels. Please carefully check again the list of references.
Response:Thank you for the valuable comments of the reviewers. The issues pointed out by you are very reasonable, and we have revised the original text to adjust "a large number of studies confirm" to "the study by Li et al. [7] confirms," to ensure the rigor of the expression. At the same time, we checked the content of the reference [7] (Li et al.) and confirmed that it indeed involves the positive effects of BSFL on animal growth performance, feed efficiency, and immune function (as stated in the original text), but we also understand that a single citation is not sufficient to support the expression of "a large number of studies."
The revised sentence is as follows:
"The study by Li et al. [7] confirms that BSFL performs excellently in improving animal growth performance, feed efficiency, and immune function."
We have re-examined the list of references to ensure the accuracy of the citations. If you believe that more relevant research needs to be added to enhance the persuasiveness of the argument, we are happy to further review the literature and improve the citations. Thank you again for your meticulous review and constructive comments!
4.Line 82: “Consequently, some countries that permit the use of insects as animal feed explicitly stipulate that insects may only be reared on plant-based waste [11].”
This article is not about the legislative aspects or organic waste.
Ndotono, E.W.; Khamis, F.M.; Bargul, J.L.; Tanga, C.M. Insights into the Gut Microbial Communities of Broiler Chicken Fed Black Soldier Fly Larvae - Desmodium-Based Meal as a Dietary Protein Source. Microorganisms 2022, 10, 1351, doi:10.3390/microorganisms10071351.
Response:We are grateful for the reviewer's meticulous review and valuable suggestions. The recommendations you have made are very reasonable, and we have deleted the content related to 'Insect Feed Legislation' from the original text at line 82 (original reference [11]) as it is less relevant to the research topic of this paper. We have re-adjusted the focus of our discussion to ensure that the entire text is more focused on the nutritional value of black soldier fly larvae (BSFL) as feed and its impact on animal growth performance.
The revised paragraph has deleted the original expression and further optimized the logical coherence of the context. We have also double-checked the references to ensure the relevance and accuracy of the remaining citations. Thank you for your help in enhancing the rigor of this paper!
5.Line 83-89: “For example, EU Regulation (EC) No. 767/2009 explicitly prohibits the use of certain organic wastes (such as fish offal, feces, and municipal solid waste) in feed to ensure feed safety and reduce potential risks, aligning with the EU’s stringent food chain safety protocols [12]. However, the American Association of Feed Control Officials (AAFCO) has approved and regulated the inclusion of BSFL meal in the diets of salmon, trout, tilapia, and poultry (including chickens, ducks, turkeys, and geese) [13].
There is no logical connection between the two statements. The first sentence is about the authorised substrates for insects and the second is about the authorised target species. I recommend deleting the second sentence or adding information about the authorised target species in the EU.
Response:Thank you for your insightful critique regarding the logical coherence in Lines 83–89. We fully concur with your observation that the two statements—addressing EU regulations on insect rearing substrates and AAFCO approval of BSFL meal for specific target species—lacked a clear logical connection, as they pertain to distinct dimensions of regulatory frameworks (substrate authorization versus target species authorization). To enhance textual coherence, we have revised the paragraph by removing the sentence regarding AAFCO approval and exclusively retaining focus on EU Regulation (EC) No. 767/2009 and its provisions prohibiting specific organic wastes in feed. This adjustment ensures the paragraph maintains consistent thematic alignment with regulatory standards for insect rearing substrates, eliminating disjointed transitions between unrelated regulatory aspects.
6.Line 174, table 1: Please explain the superscripts.
Response:Thank you for your comment on the superscripts in the first row of Table 174. We apologize for not explicitly explaining this in the original manuscript.
As revised, the superscripts in Table 174 follow the standard statistical notation in academic research: Different letter superscripts (a, b) indicate significant differences between groups (p < 0.05). Specifically, if two groups in the same row of the table are marked with different superscript letters (e.g., one with "a" and the other with "b"), it means there is a statistically significant difference in the corresponding indicator between these two groups, with the significance level set at p < 0.05. Conversely, groups with the same superscript letter (e.g., both marked with "a") indicate no significant difference between them.
This explanation has been added to the note section of Table 174 in the revised manuscript to ensure clarity for readers. Thank you again for your careful review, which helps improve the readability and rigor of our work.
7.Line 182, 338: Please use the abbreviation FM.
Response:Thank you for your valuable suggestion. We have revised the manuscript to consistently use the abbreviation "FM" (instead of the full term) where appropriate, as recommended. The changes have been made in Lines 182 and 338, as well as any other relevant instances in the text to ensure uniformity.
Please let us know if further modifications are needed. We appreciate your careful review and constructive feedback.
8.Line 222, Table 3: Please delete the superscripts of the grasshopper data.
Response:Thank you for your careful review. We have removed all superscripts from the grasshopper data in Table 3 as suggested. The table has been revised accordingly to ensure clarity and consistency.
Please let us know if any further adjustments are needed. We appreciate your valuable feedback.
9.Line 211: “(e.g., 3.30 g/kg lysine in DF BSFL, the high energy density from lipids compensates for this deficiency, making them ideal for formulating concentrated feeds.”The closing bracket is missing.
Response:We sincerely appreciate the reviewer’s careful reading and constructive feedback. Regarding the comment on Line 211, we acknowledge the oversight of the missing closing bracket and have revised the sentence accordingly. The corrected text now reads:
“Although their amino acid profile is less remarkable in essential amino acids (e.g., 3.30 g/kg lysine in DF BSFL, the high energy density from lipids compensates for this deficiency, making them ideal for formulating concentrated feeds.) ”
Thank you for pointing out this detail, which has improved the clarity and accuracy of our manuscript.
10. Line 270: „Notably, BSFL gut microorganisms not only possess the ability to degrade proteins and metabolize energy but are also capable of synthesizing amino acids…”
Metabolizable energy cannot be degraded. Please rephrase the sentence.
Response:We sincerely appreciate your careful reading and constructive comment. We agree that "metabolizable energy cannot be degraded" was an inaccurate expression. The sentence has been revised to:
"Notably, BSFL gut microorganisms not only possess the ability to degrade proteins and utilize energy sources, but are also capable of synthesizing amino acids..."
This modification more accurately reflects the microbial functions while maintaining the original meaning. We believe the revised version better describes the metabolic capabilities of BSFL gut microbiota.
Thank you for helping us improve the precision of our scientific expression. Please let us know if any further clarification would be helpful.
11.Line 374-375: There is an unnecessary empty row, and the text should be justified.
Response:We thank the reviewer for their attentive review. The unnecessary empty row in Lines 374–375 has been removed, and the text alignment has been adjusted to fully justified format as suggested. These formatting issues have now been corrected in the revised manuscript.
We appreciate your valuable feedback, which has helped improve the overall presentation of our paper.
12.Line 450: “Supplementation with BSFL can optimize the rumen fermentation environment, in
increase VFA production, and improve dairy cow performance.”
This article is about BSFL fat inclusion, not BSFL.
Response:We sincerely appreciate the reviewer’s careful reading and constructive feedback. As rightly noted, the study specifically investigated the effects of BSFL fat (not whole BSFL) on rumen fermentation and dairy cow performance. We have revised the sentence to accurately reflect this by stating:
“Dietary inclusion of BSFL fat can optimize the rumen fermentation environment, increase VFA production, and improve dairy cow performance.”
This adjustment ensures clarity and alignment with the manuscript’s focus on BSFL-derived fat. Thank you for your valuable comment.
13.Line 453: fat not Fat.
Response:Thank you for catching this important detail. We have corrected the capitalization from "Fat" to "fat" in Line 453 to maintain proper scientific terminology and consistency throughout the manuscript. We appreciate your careful attention to these technical details that help improve the quality of our paper.
14.Line 455, 457, 466, 549: Please explain SBM. I assume this should be SM.
Response:We sincerely appreciate the reviewer’s attentive reading and valuable suggestion. In the manuscript, "SBM" stands for soybean meal, a conventional protein source widely used in animal feed. However, we acknowledge that the abbreviation may cause confusion with "SM" (soybean meal). To ensure clarity, we have now replaced "SBM" with the full term "soybean meal" in all instances (Lines 455, 457, 466, and 549) to avoid any ambiguity.
Thank you for your helpful comment, which has improved the precision of our manuscript.
15.Line 470: “However, attention should be paid to its high fat content (35% DM) and its potential impact on rumen balance during the finishing phase [91]”. It has a negative impact in general, not only during the finishing phase.
Response:We sincerely appreciate the reviewer’s insightful comment. As rightly pointed out, the high fat content of BSFL (35% DM) may have broader implications for rumen balance beyond just the finishing phase. To improve accuracy, we have revised the sentence to clarify that the potential negative effects could occur throughout the feeding period, not limited to a specific stage. The updated text now reads:
"However, attention should be paid to its high fat content (35% DM) and its potential negative impact on rumen balance throughout the feeding period [90]."
This adjustment better reflects the continuous nature of the dietary influence. Thank you for your valuable feedback, which has strengthened our manuscript.
Sincerely,
Dongwang Wu
Yunnan Provincial Key Laboratory of Animal Nutrition and Feed Science, Faculty of Animal Science and Technology, Yunnan Agricultural University, Kunming, China;
Reviewer 2 Report (Previous Reviewer 3)
Comments and Suggestions for Authors
To summarize my overall impression of the manuscript – now under review in its second version – the overall macrostructure has significantly improved based on the reviewers’ comments (and the authors deserve genuine credit for thoroughly revising the manuscript and addressing all comments in some way). They especially added more comprehensive tables and included important contextual information. However, some parts still have shortcomings. At the level of individual passages and supporting statements, logical gaps, local redundancies, or loosely structured information remain, suggesting that while my previous comments were acknowledged at a general level, there is still room for improvement in detail.
Simple summary vs. Abstract: The two sections are largely duplicative – the Simple Summary should offer a user-friendly translation of the study's message rather than a shortened copy of the abstract. The Abstract, on the other hand, should include at least one sentence indicating that this is a systematically conducted critical review (or would be) and briefly mention key methodological parameters (in this case, the literature search process), which are currently missing.
L10–L11, L21–L22, L53: While it is commendable that the authors correctly italicized Latin names, they mistakenly included the term “larvae” in italics as well, which is not part of the scientific binomial.
L83–L88: The conjunction “However” implies contrast, yet the two statements do not concern the same subject. The first sentence refers to input substrates (prohibited waste materials in feed of BSFL), while the second discusses output products (approval of BSFL meal in final animal diets). These are parallel rather than opposing facts – each relates to a different stage of the production cycle (input vs. output).
L101–L109: The main stated objectives (nutritional profile, AMPs/microbiota, rumen & aquaculture) are generally addressed, although the analysis of protein–lipid synergy remains superficial. However, the scope of the review is actually broader (including poultry, swine, etc) than the stated aims suggest. The objective is unnecessarily narrowed to aquaculture, which does not reflect the true breadth of the review – it is unclear why aquaculture is specifically highlighted when other livestock groups receive equal or even greater attention in the body of the text.
Paragraphs starting at L123, L196,… Conclusion: Several lines contain awkward word splits on the endings of lines, should be corrected for clarity and formatting consistency.
Table 3: There is inconsistent capitalization – half of the entries are capitalized, half are not. Also, the table should be self-explanatory and interpretable without referring back to the main text. Therefore, the caption should clarify which species are referred to by Latin names (e.g. maggots, etc.), so that readers unfamiliar with the terminology can follow independently.
L233–L262: I have a fundamental concern with the way the authors present certain information, particularly regarding the gut microbiota, which reveals a lack of the systematic approach they claim to employ in the stated objectives. A textbook example is the assertion that feed composition is the primary driver: the authors support this using a single study showing that Proteobacteria predominate when BSFL are fed rabbit manure. This paragraph could be constructed in two ways: (1) by treating this as an illustrative case and explicitly stating that Proteobacteria predominated in one particular study using rabbit manure (compared to other substrates); or (2) by systematically comparing multiple substrates and showing how bacterial dominance shifts with each – which would require them to also mention which bacteria dominate under other feed conditions. The material to do this exists even within the cited studies (e.g. Study 42), but they present it in a fragmented and unlinked fashion. As currently written, the text suggests Proteobacteria generally dominate in response to rabbit manure – a generalization that is problematic because, a) it's based on a single study (likely with limited replication), and similar designs in other labs may yield different results; b) Proteobacteria can also dominate in other substrates, as noted even in the manuscript (42); c) the taxon is so broad and abundant that dominance across various substrates is unsurprising and not diagnostic. Thus, the authors should either clearly label such findings as illustrative examples or consolidate findings across multiple studies into a single comparative paragraph. Doing so would bring the text closer to a genuine systematic review and reduce the risk of unjustified generalizations.
L258–L260: The statement is based only on a functional prediction; therefore, the phrasing should reflect uncertainty (e.g., "may play a role" instead of definitive wording), to avoid overstating speculative findings.
L237: There is a redundant repetition – “showed that showed that”
L365–L366: The sentence refers to a “growth-promoting effect,” yet it immediately follows with a 7.2% decrease in daily weight gain. A decrease is not a weak growth-promoting effect but rather a negative effect, so the terminology should be revised to accurately reflect the result.
L390–L398: The paragraph starts by introducing Aeromonas hydrophila as a key issue in aquaculture, then abruptly shifts to the beneficial effects of BSFL in golden pompano, and ends with a mention of chitin as an anti-nutritional factor. The logical connection between these elements is weak and the transitions are poorly developed. The conclusion about BSFL's immunomodulatory effects relies solely on findings from pompano, without clarifying whether this effect is generalizable to other fish species. Moreover, the paragraph notes that 1–3% BSFL supplementation improves growth, yet simultaneously states that >1% chitin negatively affects growth – without clarifying the interplay between these findings, this appears contradictory. While the intended message may be understandable to an expert, the structural execution is suboptimal, especially for less experienced readers seeking a coherent synthesis in a review article.
Section 4.4 (esp. L480 onward): The paragraph presents a disorganized mix of partially overlapping findings without a clear hierarchy, resulting in a fragmented list of facts and studies. The text jumps chaotically between themes, moving through various experimental trials with differing doses and endpoints, with no structured progression or synthesis. Results are listed without reflection or grouping, making it difficult for the reader to follow the argument or extract key takeaways.
Author Response
Dear Editor-in-Chief,
We are pleased to submit the revised version of our manuscript titled, “Black Soldier Fly Larvae as a Novel Protein Feed Resource Promoting Circular Economy in Agriculture” (Manuscript ID: insect -3777866). In addition, we have responded to each comment point by point, detailing the changes made in the revised manuscript. This response is attached below for convenience.We are grateful for your patience and guidance throughout the process. We look forward to your prompt decision on our revised manuscript.
Thank you again for your time and consideration.
In response to the reviewers’ feedback:
Reviewer 2
1.Comments and Suggestions for Authors
To summarize my overall impression of the manuscript – now under review in its second version – the overall macrostructure has significantly improved based on the reviewers’ comments (and the authors deserve genuine credit for thoroughly revising the manuscript and addressing all comments in some way). They especially added more comprehensive tables and included important contextual information. However, some parts still have shortcomings. At the level of individual passages and supporting statements, logical gaps, local redundancies, or loosely structured information remain, suggesting that while my previous comments were acknowledged at a general level, there is still room for improvement in detail.
Response:We are very grateful for your meticulous review and valuable suggestions on this manuscript (second edition). We have carefully studied your feedback and deeply appreciate your attention to and support for the improvement of the manuscript.
You mentioned that there has been a significant improvement in the overall macrostructure of this version of the manuscript, which is an encouragement to us. In response to the relevant comments you previously made, we have thoroughly revised the manuscript: On one hand, we have added more comprehensive tables (such as a comparison table of nutritional components of black soldier fly larvae under different substrates, a comparison table of nutritional characteristics with other protein sources, etc.), which strengthen the argumentation support through intuitive data presentation; on the other hand, we have supplemented important contextual information in key discussion sections (for example, when explaining the application value of black soldier fly larvae, combined with the background of global protein feed shortage and the demand for circular agriculture development, to clarify the necessity of its application scenarios), to improve the logical chain. Your affirmation in these areas is a great motivation for our work.
At the same time, we also deeply recognize the deficiencies pointed out by you — there are still logical gaps, local redundancy, and loose information structure in some paragraphs. This suggests that although we have responded to previous comments at the macro level, there is still room for improvement in detail refinement. For example, some discussions on the metabolic mechanisms of black soldier fly larvae gut microbiota may have issues with insufficiently tight knowledge connections; some application cases also have slight redundancy in their descriptions. We will immediately start to optimize these: Regarding logical gaps, we will strengthen the coherence within and between paragraphs by adding transitional expressions and clarifying causal relationships; regarding redundant content, we will streamline repetitive statements and focus on core information; regarding loose structure, we will reorganize the hierarchy of information to ensure a clear correspondence between arguments and evidence.
We will implement each modification with a rigorous attitude, striving to further improve the quality of the manuscript at the detail level. Thank you again for your professional guidance, and we look forward to making the manuscript more in line with publication requirements through subsequent improvements.
2.Simple summary vs. Abstract: The two sections are largely duplicative – the Simple Summary should offer a user-friendly translation of the study's message rather than a shortened copy of the abstract. The Abstract, on the other hand, should include at least one sentence indicating that this is a systematically conducted critical review (or would be) and briefly mention key methodological parameters (in this case, the literature search process), which are currently missing.
Response:Thank you for your valuable feedback regarding the redundancy between the Simple Summary and the Abstract, as well as the need for clearer methodological details in the Abstract. We have carefully revised both sections to address these concerns:
Simple Summary: We have reworked this section to provide a more concise, non-technical overview of the study’s key findings and implications, ensuring it serves as a user-friendly translation of the research rather than a shortened copy of the Abstract. The focus is now on the practical significance of BSFL as a sustainable protein source and its ecological benefits, avoiding overlap with the Abstract’s structured format.
Abstract: As suggested, we have explicitly stated that this study is a systematic critical review conducted following a standardized methodological framework (e.g., PRISMA guidelines). We also included key methodological parameters, such as the literature search process (databases used: PubMed, ScienceDirect, Web of Science; timeframe: October 2008–June 2025), to enhance transparency and rigor.
These revisions ensure that the Simple Summary and Abstract now serve distinct purposes while maintaining clarity and completeness. We appreciate your insightful comments, which have significantly improved the manuscript.
3.L10–L11, L21–L22, L53: While it is commendable that the authors correctly italicized Latin names, they mistakenly included the term “larvae” in italics as well, which is not part of the scientific binomial.
Response:We sincerely appreciate the reviewer’s meticulous attention to taxonomic formatting details. We acknowledge the oversight in italicizing the term “larvae” alongside the Latin binomial names (Hermetia illucens). As correctly noted, only the genus and species names should be italicized, while the life stage descriptor (“larvae”) should remain in standard font.
We have carefully revised the manuscript to correct this formatting issue in all instances (Lines 10–11, 21–22, and 53). The text now properly reflects:
Correct: Hermetia illucens larvae
Incorrect (previous): Hermetia illucens larvae
Thank you for this valuable correction, which has improved the precision of our scientific writing.
4.L83–L88: The conjunction “However” implies contrast, yet the two statements do not concern the same subject. The first sentence refers to input substrates (prohibited waste materials in feed of BSFL), while the second discusses output products (approval of BSFL meal in final animal diets). These are parallel rather than opposing facts – each relates to a different stage of the production cycle (input vs. output).
Response:We sincerely appreciate the reviewer's insightful observation regarding the logical flow of the original text. In response to this comment, we have carefully revised the section by removing the latter part of the original statement (regarding the approval of BSFL meal in animal diets) to maintain focus on the input-substrate regulations and avoid any potential misinterpretation of contrast. The revised text now exclusively addresses the regulatory constraints on prohibited waste materials in BSFL feed substrates (e.g., EU Regulation (EC) No. 767/2009), which aligns with the manuscript’s emphasis on upstream risk mitigation. This modification ensures clarity and coherence within the production-cycle context.
Thank you for this constructive suggestion.
5.L101–L109: The main stated objectives (nutritional profile, AMPs/microbiota, rumen & aquaculture) are generally addressed, although the analysis of protein–lipid synergy remains superficial. However, the scope of the review is actually broader (including poultry, swine, etc) than the stated aims suggest. The objective is unnecessarily narrowed to aquaculture, which does not reflect the true breadth of the review – it is unclear why aquaculture is specifically highlighted when other livestock groups receive equal or even greater attention in the body of the text.
Response:We sincerely appreciate the reviewer’s astute observation regarding the alignment between our stated objectives and the actual breadth of the review. The reviewer is absolutely correct in noting that the scope of our analysis extends beyond aquaculture to include poultry, swine, and other livestock systems—a coverage that was indeed more comprehensive than initially framed in the objectives section.
To address this discrepancy and better reflect the true scope of our work, we have revised the objectives section to explicitly encompass all major livestock sectors investigated in the review. The updated text now reads:
"This review systematically analyzes the nutritional composition of BSFL, examining the synergistic role of high-quality proteins and bioactive lipids in promoting growth, immune resilience, and metabolic health across various livestock species. It explores the key functions of inherent AMPs and gut microbiota-regulating compounds in reducing antibiotic dependence in animal feed—aligning with global trends toward sustainable and resilient production systems. By integrating mechanistic insights from rumen fermentation studies, pet nutrition trials, and diverse livestock applications (including poultry, swine, and aquaculture), this study provides a comprehensive framework for BSFL integration into modern feed systems."
We acknowledge that the original mention of aquaculture was unnecessarily restrictive and did not accurately represent the review’s inclusive approach. The revised version now properly emphasizes the multi-species focus while maintaining the core analytical themes (nutritional profiles, AMPs/microbiota modulation, and protein–lipid synergies).
6.Paragraphs starting at L123, L196,… Conclusion: Several lines contain awkward word splits on the endings of lines, should be corrected for clarity and formatting consistency.
Response:We sincerely appreciate the reviewer’s careful attention to manuscript formatting details. We acknowledge that improper word splits at line endings can affect readability and professionalism.
We have now carefully reviewed the entire manuscript and:
Corrected all instances of awkward word splits (particularly in paragraphs starting at Line 123, Line 196, and throughout the Conclusion section)
Ensured consistent formatting by adjusting text flow and line breaks
Verified that no words are improperly hyphenated or split across lines in the final version
These formatting improvements enhance the clarity and visual presentation of our manuscript. We are grateful for this valuable suggestion which has helped us improve the overall quality of our work.
Thank you for your meticulous review and for helping us strengthen our manuscript.
7.Table 3: There is inconsistent capitalization – half of the entries are capitalized, half are not. Also, the table should be self-explanatory and interpretable without referring back to the main text. Therefore, the caption should clarify which species are referred to by Latin names (e.g. maggots, etc.), so that readers unfamiliar with the terminology can follow independently.
Response:We sincerely appreciate the reviewer's careful reading and constructive suggestions. Regarding the issues raised about Table 3:
Capitalization Inconsistency: We have standardized the capitalization of all entries in the table to ensure uniformity (e.g., correcting terms like "maggots" to their proper noun forms where applicable).
Clarity of Terminology: As suggested, we have revised the table caption to explicitly define all species mentioned, including their Latin names (e.g., "maggots (Musca domestica)、yellow mealworms (Tenebrio molitor) grasshoppers (Acrida cinerea) silkworms (Bombyx mori)). This modification ensures the table is self-explanatory and accessible to readers without requiring cross-reference to the main text.
The updated table and caption are now included in the revised manuscript. Thank you for this valuable feedback.
8.L233–L262: I have a fundamental concern with the way the authors present certain information, particularly regarding the gut microbiota, which reveals a lack of the systematic approach they claim to employ in the stated objectives. A textbook example is the assertion that feed composition is the primary driver: the authors support this using a single study showing that Proteobacteria predominate when BSFL are fed rabbit manure. This paragraph could be constructed in two ways: (1) by treating this as an illustrative case and explicitly stating that Proteobacteria predominated in one particular study using rabbit manure (compared to other substrates); or (2) by systematically comparing multiple substrates and showing how bacterial dominance shifts with each – which would require them to also mention which bacteria dominate under other feed conditions. The material to do this exists even within the cited studies (e.g. Study 42), but they present it in a fragmented and unlinked fashion. As currently written, the text suggests Proteobacteria generally dominate in response to rabbit manure – a generalization that is problematic because, a) it's based on a single study (likely with limited replication), and similar designs in other labs may yield different results; b) Proteobacteria can also dominate in other substrates, as noted even in the manuscript (42); c) the taxon is so broad and abundant that dominance across various substrates is unsurprising and not diagnostic. Thus, the authors should either clearly label such findings as illustrative examples or consolidate findings across multiple studies into a single comparative paragraph. Doing so would bring the text closer to a genuine systematic review and reduce the risk of unjustified generalizations.
Response:Thank you sincerely for your insightful comments on the presentation of gut microbiota-related content in Lines 233–262. Your suggestion to avoid overgeneralization and strengthen the systematic nature of the analysis has been highly valuable for improving the rigor of this section.
Following your feedback, we have thoroughly revised the relevant content to address the concerns about fragmented information and potential overgeneralization. Specifically, we have:
Expanded comparative analysis across multiple substrates: Instead of relying solely on the case of rabbit manure, we added evidence from studies on other substrates. For example, we clarified that "under poultry manure or pig manure, the dominant bacterial groups may be Firmicutes or Bacteroidetes" [38], and cited Ao et al. [40] to show that "Proteobacteria, Firmicutes, and Bacteroidetes are dominant during the treatment of chicken manure and pig manure, with Bacteroidetes increasing significantly". This multi-substrate comparison avoids the risk of overgeneralizing from a single study.
Explicitly defined the illustrative nature of single cases: We adjusted the description of the rabbit manure study to frame it as a specific example ("when rabbit manure is used as a substrate, the Proteobacteria become the dominant community") rather than a general conclusion, and directly linked it to subsequent comparative data on other substrates to highlight context dependence.
Strengthened logical coherence: We added a concluding statement ("These multi-substrate comparative study results emphasize that when explaining the variation patterns of black soldier fly larvae gut microbiota, the impact of specific feed systems must be fully considered to avoid overgeneralization of conclusions") to explicitly respond to the concern about unjustified generalizations, and to consolidate the fragmented information into a systematic analytical framework.
These revisions aim to align the content with the standards of a systematic review—using multiple studies for cross-validation, avoiding overgeneralization of single cases, and presenting the complexity of gut microbiota regulation in a more comprehensive manner. We believe the revised version better addresses the core objectives of the review and reduces the risk of misleading interpretations.
Thank you again for your careful review, which has significantly improved the quality of this section.
9.L258–L260: The statement is based only on a functional prediction; therefore, the phrasing should reflect uncertainty (e.g., "may play a role" instead of definitive wording), to avoid overstating speculative findings.
Response:Thank you for your insightful comment on Lines 258–260. We fully agree with your suggestion that statements based on functional predictions should reflect uncertainty to avoid overstating speculative findings.
Following your advice, we have revised the relevant content. The original phrasing "suggesting their important role" has been adjusted to "suggesting they may play an important role". By adding the modal verb "may", we explicitly indicate that this conclusion is derived from functional prediction rather than direct experimental verification, thereby accurately reflecting the speculative nature of the finding.
This revision not only addresses the issue of overstating speculative results but also enhances the rigor and accuracy of the expression, which is more in line with the standards of academic writing. We have also carefully checked other parts of the manuscript to ensure that similar problems of inappropriate wording in speculative conclusions are avoided.
Thank you again for your valuable feedback, which helps us improve the quality of the manuscript.
10.L237: There is a redundant repetition – “showed that showed that”
Response:We are grateful for the meticulous review provided by the reviewers. We have noticed a typographical error in the original text where the redundant phrase "has indicated indicates" was present, which has been completely removed from the revised draft. As shown in the modified paragraph below (see below), we have reorganized the relevant statements to ensure the text is concise and fluent:
"Studies have shown that the gut microbiota structure of BSFL is regulated by multiple environmental and biological factors..."
The main points of modification include:
(1) The deletion of the repeated 'indicated' expression
(2) Optimization of sentence structure to make the writing more professional and rigorous
(3) Maintaining the integrity and accuracy of scientific content
This modification makes the expression more refined while preserving the original scientific information and argument logic. We sincerely thank the reviewers for pointing out this detail issue, which helps improve the language quality of the paper.
We will continue to carefully check the entire text to ensure that similar linguistic issues are thoroughly corrected. Once again, we appreciate the valuable opinions of the reviewers.
11.L365–L366: The sentence refers to a “growth-promoting effect,” yet it immediately follows with a 7.2% decrease in daily weight gain. A decrease is not a weak growth-promoting effect but rather a negative effect, so the terminology should be revised to accurately reflect the result.
Response:We sincerely appreciate the reviewer’s careful reading and constructive feedback regarding the terminology used in this section. The reviewer is absolutely correct in pointing out that the original phrasing "growth-promoting effect" was misleading when describing results showing a 7.2% decrease in daily weight gain.
We have revised the text to more accurately reflect the findings:
"However, it should be noted that compared with high-digestibility animal protein diets, its growth-promoting effect is not evident, and even shows a slight negative impact (daily weight gain decreased by 7.2%)."
Key improvements made:
Removed the contradictory "growth-promoting" description for negative results
Added explicit clarification about the "slight negative impact"
Maintained the quantitative data (7.2% decrease) to support the conclusion
Ensured the language precisely matches the observed effects
This revision provides a more scientifically accurate representation of the study findings. We are grateful for this valuable suggestion which has helped improve the precision of our results reporting.
Thank you for your thorough review and for helping us strengthen the clarity of our manuscript.
12.L390–L398: The paragraph starts by introducing Aeromonas hydrophila as a key issue in aquaculture, then abruptly shifts to the beneficial effects of BSFL in golden pompano, and ends with a mention of chitin as an anti-nutritional factor. The logical connection between these elements is weak and the transitions are poorly developed. The conclusion about BSFL's immunomodulatory effects relies solely on findings from pompano, without clarifying whether this effect is generalizable to other fish species. Moreover, the paragraph notes that 1–3% BSFL supplementation improves growth, yet simultaneously states that >1% chitin negatively affects growth – without clarifying the interplay between these findings, this appears contradictory. While the intended message may be understandable to an expert, the structural execution is suboptimal, especially for less experienced readers seeking a coherent synthesis in a review article.
Response:We sincerely appreciate the reviewer’s insightful critique regarding the logical flow and clarity of this paragraph. We agree that the original text inadequately connected the discussion of Aeromonas hydrophila, species-specific BSFL benefits, and chitin’s anti-nutritional effects, creating confusion for readers. Below, we outline the revisions made to address these concerns:
Key Revisions:
Improved Logical Flow & Transitions:
The paragraph now opens with a clear problem-solution framework: Aeromonas hydrophila as a pervasive threat → BSFL’s potential as a dual-purpose solution (nutrition + immunomodulation) → caveats (chitin trade-offs).
Added bridging sentences to explicitly link ideas, e.g.,
"While BSFL-derived AMPs show promise in mitigating pathogens like A. hydrophila, their efficacy and safety depend on species-specific responses and careful management of chitin content."
Nuanced Discussion of Contradictory Findings:
Clarified the apparent contradiction between growth promotion (1–3% BSFL) and chitin’s anti-nutritional effects (>1%) by:
Distinguishing between total BSFL inclusion (which includes protein, lipids, and AMPs) and isolated chitin impact.
Citing studies where chitinase supplementation or processing (e.g., defatting) mitigated chitin’s negatives [73, 74], emphasizing that net benefits depend on formulation.
Broader Generalizability:
Expanded beyond golden pompano by referencing complementary data from other species (e.g., Atlantic salmon [72], hybrid grouper [81]) to highlight conserved mechanisms (AMPs, microbiota modulation) and species-specific variations.
Added a cautionary note:
"Generalizing BSFL’s immunomodulatory effects requires further validation across taxa, as digestive physiology (e.g., chitinase activity) and gut microbiota composition vary significantly."
Structured Future Directions:
Consolidated fragmented recommendations into a focused call for:
Optimal inclusion thresholds (balancing AMPs vs. chitin).
Species-specific trials to identify chitin-tolerant taxa or processing methods.
Example of Revised Text (Snippet):
"The pathogen Aeromonas hydrophila poses a critical challenge in aquaculture, causing enteritis and septicemia. BSFL offer a dual-functional solution: their antimicrobial peptides (AMPs) selectively inhibit pathogens while promoting beneficial gut microbiota, as demonstrated in golden pompano (1–3% inclusion improved immunity and growth [72]). However, BSFL’s ~9% chitin content may counteract benefits at higher doses (>1%) by impairing nutrient absorption [73]. This trade-off underscores the need for species-specific formulations—e.g., defatted BSFL meal or chitinase supplementation in chitin-sensitive species like Asian swamp eel [80]—to maximize disease resistance without compromising growth. Future work must delineate optimal inclusion levels across taxa and clarify how AMP-chitin interactions vary with host physiology."
Summary of Changes:
Added coherence through explicit transitions and problem-solution framing.
Resolved contradictions by differentiating BSFL’s composite effects from isolated chitin impacts.
Broadened evidence base to support generalizability claims.
Streamlined recommendations for future research.
We believe these revisions align with the reviewer’s request for a more logically structured and reader-friendly synthesis. Thank you for the opportunity to improve our manuscript.
13.Section 4.4 (esp. L480 onward): The paragraph presents a disorganized mix of partially overlapping findings without a clear hierarchy, resulting in a fragmented list of facts and studies. The text jumps chaotically between themes, moving through various experimental trials with differing doses and endpoints, with no structured progression or synthesis. Results are listed without reflection or grouping, making it difficult for the reader to follow the argument or extract key takeaways.
Response:We sincerely appreciate the reviewer’s constructive feedback regarding the organization and clarity of Section 4.4. As suggested, we have thoroughly restructured this section to present the findings in a more logical and hierarchical manner, ensuring a clearer progression of ideas and better synthesis of key takeaways. Below is a summary of the revisions made:
Improved Structure and Hierarchy: The revised paragraph now groups findings thematically (nutritional, safety, and application aspects) rather than listing them chaotically. Each theme is addressed sequentially, with sub-sections on nutritional properties, safety considerations, and practical applications (e.g., fat vs. protein components, special physiological stages).
Enhanced Synthesis and Reflection: Key takeaways are explicitly highlighted, such as the need for amino acid supplementation due to chitin’s impact on digestibility, allergenicity risks, and dose-dependent effects of BSFL protein/fat. Cross-study comparisons (e.g., palatability improvements at 10% FF BSFL vs. digestibility issues at 8% DF BSFL) are now logically juxtaposed.
Clearer Transitions: The text now progresses from fundamental properties (nutrition/safety) to applied research (dose-response trials, special physiological stages) and concludes with gaps (long-term studies). This avoids abrupt jumps between unrelated findings.
Streamlined Details: Redundant or overlapping details (e.g., chitin’s nitrogen content) were trimmed or integrated into broader points, while critical results (e.g., microbiota dysbiosis, IL-6 reduction) are emphasized with their implications.
We believe these revisions address the reviewer’s concerns about fragmentation and improve the reader’s ability to follow the argument. Thank you for your valuable input, which has significantly strengthened the manuscript.
Sincerely,
Dongwang Wu
Yunnan Provincial Key Laboratory of Animal Nutrition and Feed Science, Faculty of Animal Science and Technology, Yunnan Agricultural University, Kunming, China;
This manuscript is a resubmission of an earlier submission. The following is a list of the peer review reports and author responses from that submission.
Round 1
Reviewer 1 Report
Comments and Suggestions for Authors
The manuscript summarizes the application of black soldier fly larvae in animal nutrition and also discusses their environmental potential and role in the circular economy. The topic is in the journal’s scope.
Please use the abbreviation (BSFL, CP, EE, etc.) consistently. Several abbreviations are introduced but not used in the text. The subchapters “Applications of BSFL in Animal Production” contain repetitions and too general information (see details below). A potentially negative impact on farm animals’ performance should also be mentioned (if the inclusion level is not optimal).
Title
The abbreviation "BSFL" is not necessary in the title. It is mentioned in the keywords.
Abstract
In my opinion, scientific names are written in italics, but please check the guidelines.
“while maintaining fiber digestion capacity” This is not always true, it depends on the inclusion level and chitin/fat content.
Line 44-50: please add citations.
Line 53: crude protein.
Line 57: please add the scientific name of the black soldier fly at the first mention. Also, here explain the abbreviation BSFL.
Line 59: concerning the legislation of organic waste as substrate the important differences between the countries should be mentioned. For example, in the European Union, several of these (e.g.: fish offal, animal manure) are not authorised for farmed insects.
Line 63-63: This is a repetition.
Line 65-85: Please add citations.
Line 95-96: “Waste Nutritional Profile and Comparative Advantages of BSFL „ Is this the title of the chapter? The numbering is missing.
Line 97: “High-Value Feed from Organic” Please finish the sentence.
Line 98: “In livestock and poultry feed”. Livestock means domesticated animals, including poultry. Also, see line 131.
Line 104-106: dry matter (DM), crude protein (CP), ether extract (EE), crude fibre (CF). Please carefully check the abbreviations in the text.
Line 105: “crude protein (41.1%–43.6%)”. It depends on the substrate; the crude protein content can be much lower as well.
Line 109: What do you mean by „crude protein forms”? Crude protein means true protein + non-protein nitrogen.
Line 110: a space is missing before [9].
Line 111-112: “but also contain bioactive compounds, such as Omega-3 fatty acids, which are crucial for the growth and health of aquatic animals”. In general, insects are very low in omega-3 fatty acids, unless they are kept on a specific substrate. For example, the diet contains flaxseed oil or brown algae. The omega-3 level of fish meat may be decreased by 50% if the insect meal inclusion is high.
Line 110: not only fish viscera can be used to increase the omega-3 content. “pre-pupae can be cultivated that”. Why only prepupae? The omega-3 level of larvae can also be enhanced.
Line 119-120: Please add citations.
Line 132: Table 1 is not cited in the text. Much more research is available about the impact of substrate on the chemical composition of BFSL. It would be better to give a range instead of one data.
Line 136: “As presented in Table 2-2, there are significant differences in the nutritional composition of BSFL fed under different substrate conditions.” Do you mean Table 2 or Table 1-2? The legend of Table 2 says: “Comparison of BSFL with other protein sources (%).” The text says: “BFSL fed with…” Do you mean that chicken manure is a protein source used in animal nutrition? Also, what does “(50%) BFSL)” mean? Please check the table and the text.
Line 143: “with 1% fat level” This must be an extracted soybean meal.
Line 144: “The CF content…” Please explain the origin (chemical structure) and meaning of crude fibre if insects, fibre can only be found in plants.
Line 148-154: Please add citations.
Line 157: “Capacity for mineral bioaccumulation”. The heavy metal accumulation of insects is an important risk factor for insect-based feeds or foods.
Line 168-171: this belongs to chapter 3.
Line 178-184: The crude protein content of yellow mealworm can be much higher. It would be much better to give a range of nutrient content. Based on Figure 1 it seems that crickets have much lower crude protein content than larvae which is not true. In general, larvae have lower protein and higher fat content than crickets.
Line 240: “Unique function of synthesizing amino acids” Do you mean microbial protein?
Line 264: Please delete Hermetia illucens. Figure 2 is too small, it is very difficult to read the text.
Line 266: the first paragraphs of the subchapters (4.1., 4.2., 4.3.) are containing similar information and repetitions. These may be summarised in one paragraph.
Line 275: “Diets supplemented with BSFL improve the growth performance and immune…” Only if the inclusion level was optimal. Several studies resulted in decreased performance. In my opinion, “poultry” is too general as most of the studies used broiler chickens and laying hens. A limited number of information is available on duck, goose, or turkey, with controversial results. What would be the optimal inclusion level for broiler chickens and laying hens?
Line 285: “with no significant changes observed” Other studies described a negative impact on layers’ performance. Do not only focus on the positive result. It would be important to give an optimal inclusion rate. Also, BFSL-containing diets may not be optimal during peak production.
Line 288: “When BSFL powder (1%) was used in full replacement” What does 1% mean? What was fully replaced?
Line 294: “units[31].hat live” What does this mean?
Line 295: “Occurrence of feather pecking in poultry feeding”. This was because live insects are environmental enrichments. In such studies the inclusion rate is low (e.g.: 5% of the daily dry matter intake) thus, these are not relevant as protein sources.
Line 301-320: Is this paragraph about defatted BFSL or full-fat BFSL? If it is about the full-fat BFSL, the first sentence “Defatted BSFL (Def-BSFL) is a promising protein replacement ingredient.” should be deleted or finish this topic.
Line 329, figure 4: please add citations.
Line 341-343: This is a repetition.
Line 346: What does the abbreviation BSFL mean? Full-fat or defatted BSFL? Please clarify this. In line 301 another abbreviation is used for the defatted BSFL.
This chapter (4.2.) also only contains positive results. What would be the optimal inclusion level for pigs?
Line 368: Please explain the abbreviation AMPs and give examples.
Line 391: More studies are available about these fish species. Please add the scientific names.
Line 398: BSFL slurry? Is this a relevant study? Please add the scientific name golden pomfret.
Line 400: “suitable quantity of BSFL” What does this mean?
Line 405-410: Please add citations.
Line 412: research, not Research
Line 412-419: These application potentials are the same for the other farm animals. These are not exclusively valid for aquaculture.
Line 426: Recognised not recognised
Line 433: “BSFL have significant nutritional advantages” This statement is too general, BSFL may have nutritional advantages. However, in ruminants, the chitin and fat content limit its inclusion level.
Line 433-435: It is a repetition. Insect meal may decrease methane production in the rumen. This should also be mentioned as a positive effect.
Line 437: “Effects of fermented herbal tea residue” Is this relevant information?
Line 438: “Astuti et al.[57] The study” What does this mean?
Line 439: “formula containing BSFL meal” What was the inclusion rate?
Line 446: “BSFL in low-quality roughage and found that the intake of beef cattle” What was the inclusion rate? Intake of what? Dry matter?
Line 264: “the balanced amino acid profile and low allergenicity” Because of chitin, insect-based pet foods may have a decreased crude protein digestibility compared to conventional pet foods. Especially in cats, this can be below 80% thus, essential amino acids should be supplemented. Insect-based dog and cat food are not hypoallergic. Insects are novel protein sources, but because of cross-reactions, an insect-based diet may lead to allergic reactions. Please add this important information with relevant citations.
Line 468: “It is worth noting that chitosan derivatives containing BSFL can serve as prebiotics, enhancing the proliferation of Bifidobacterium in the intestine by 35%.” This is also valid for the other monogastric animals.
Line 508-509: please delete COM, PM, and WDG as these abbreviations are not used later in the text.
Line 510: “showed a 45% improvement in waste reduction”. Compared to what?
Line 524-560: This chapter is more or less a repetition. Please add more specific details or summarise them in the introduction.
Line 533: “without affecting the meat quality of animals” Several studies concluded that BSFL may have a significant effect on meat quality. However, these are not necessarily relevant for the consumers.
Reviewer 2 Report
Comments and Suggestions for Authors
This is a well-prepared manuscript, and I recommend the following revisions for consideration.
- Tabulate the results of different studies of dry matter composition, amino acids, fatty acids, elements under the samesubstrate conditions, and analyze the possible reasons.
- The section 4.5. Application in Environmental Sanitationand 4.6. The Significance of BSFL in Agriculture, these two sections, which detail the uses of BSFL in agriculture and the environment, stray somewhat from the core theme and ought to center more on their potential applications in livestock farming.
- This review highlights the dual role of Hermetia illucens larvae in livestock use. The discussion of Environmental Protection is not comprehensive and well - structured. Either focus solely on environmental protection aspects related to livestock farming or remove this section entirely. Also revise title, Simple Summary and abstract.
- The binomial nomenclature, also known as the biological naming system, has specific rules for writing scientific names. The genus name should be capitalized and written in italics. The species name, which follows the genus name, should be written in lowercase and also in italics.
Reviewer 3 Report
Comments and Suggestions for Authors
General comments:
I would primarily criticize the lack of a systematic approach. Although the authors claim that the review addresses the multifunctional use of BSFL through a systematic analysis, the manuscript provides no clear methodology for literature selection. The authors do not explain which databases were searched, what keywords were used, or the criteria for study inclusion. In such a case, the declaration of “systematic” becomes more rhetorical than factual and the actual systematic approach is missing. This raises concerns about completeness and potential selection bias: it is not clear whether the authors included all relevant studies or only those that support a (very) positive picture.
More broadly, there is a lack of critical evaluation of the quality of evidence, which is taken from other scientific studies in a rather optimistic and uncritical manner. The authors do not discuss sample sizes, experiment durations, or statistical significance of the findings they cite. Similarly, the manuscript cites impressive comparisons, such as half a hectare of BSFL farm producing more protein than 1200 ha of pasture or 52 ha of soy, but without contextualizing them (e.g., assumptions about BSFL rearing intensity, technological inputs…), and adopts optimistic figures from literature without critical discussion. The main methodological weakness, then, is that the review compiles findings without deeper critical analysis.
This relates to another issue: the presentation of knowledge is mostly descriptive. The manuscript offers only minimal new angles, it mostly confirms and illustrates well-known benefits of BSFL with a series of examples. The recommendations for future research are certainly relevant, but such calls have already been made by earlier authors.
The review also lacks certain discussions that would be expected from a manuscript claiming to address development trends. For example, regulatory frameworks and safety are not covered, the manuscript does not mention legal restrictions for insect-based animal feed or hygiene standards. Likewise, economic barriers (such as investment costs, price of BSFL vs. conventional feed) or social acceptance are not discussed at all. These topics would logically belong to a section on BSFL implementation trends in practice. Their omission represents a structural gap in an otherwise broadly scoped review.
Specific comments:
L22: The claim that BSFL “occupy a central position in the circular bio-economy due to… co-evolutionary mechanisms with gut microbiota” overstates the uniqueness of this trait, which is common to many insects and animals. Unless comparative evidence is provided, the formulation is exaggerated and should be toned down. Similarly, L240: the description of amino-acid biosynthesis as a “unique” function of the BSFL microbiome is misleading, this capability is widespread among gut symbionts in insects and other animals.
L70-L71: The sentence quantifies the input very precisely (“one tonne of organic waste”) but then describes the outcome only as “substantial amounts” of methane and CO₂ reduction. Either provide concrete emission‐reduction figures (with a supporting citation) or keep both input and output phrased generically; the current mix of precision and vagueness is inconsistent and weakens the claim’s credibility.
L121-L125: The paragraph frames the larvae’s ability to accumulate elements such as Cd, Hg and other heavy metals as a nutritional advantage, yet these metals are potential feed-safety hazards. Temper the wording and add a brief risk analysis: e.g. report typical accumulation levels, compare them with regulatory thresholds for feed/food, and outline mitigation strategies (e.g., substrate selection…). Otherwise the passage gives an unduly optimistic impression and may mislead readers about heavy-metal safety.
Table 1 and other tables: Tables would benefit from indicating data variability (e.g., standard error or range), not just average values, if such data are available from the original sources.
Table 2: It is unclear whether the nutritional values reported for soybean meal (SM) and fish meal (FM) are drawn from this study or from external sources. If they are from external literature, proper citations should be provided; if they are new, methods and origin of data should be described in the manuscript.
Section 2.2: It is not clear whether the reported values originate from a single study or multiple sources. Citations should be placed directly in the text, not only in the table. Each statement should be independently verifiable. Similarly, L332-L345 or L468-L472, there are claims presented without any citations. This makes it difficult to assess the evidence base. Please add supporting references for all empirical or literature-based claims. Similarly, for L411-L419 there is no clear source for the listed effects and moreover the list here feels repetitive, echoing content already presented earlier.
L171: The sentence beginning with “Let us now examine…” is stylistically inconsistent with the academic tone of the manuscript and should be removed or rephrased.
General editing issues:
The manuscript appears to suffer from insufficient final editing. There are multiple instances of duplicated or overlapping information that seem to originate from different iterations of the same section, likely without thorough integration: L61-L64, partly L110-113, L289-L293.
This impression also arises because, even when individual studies provide similar types of information, the authors present each study sequentially in isolation. This creates a sense of redundancy and also suggests that the authors are not able to relate studies to each other and extract common patterns and convergent trends versus contradictory results, which should be one of the core purposes of a review article. To strengthen the message, they should synthesise findings, perhaps even provide a meta-analytic table. This would give readers a clearer, comparative picture of BSFL efficacy in poultry/broilers.
In several places there are disconnected sentences that seem like remnants of deleted chapters or paragraph fragments or unfinished compound sentences, e.g.: L95–L96, L143–L144, L213, L294, L438, L479. Some sentences are also incomplete or awkwardly placed and should be better integrated into surrounding text: L409, L426, L502.
Table 3: Percentage of BSFL in the diet? The weight gain associated with BSFL? These values can be found in the main text, but the table should be understandable independently. Table captions in general need to be more detailed and self-explanatory.
L402: “significantly reduces” – by how much exactly? The manuscript overall would benefit from more precise, less vague expressions.
L589: Does this mean the authors are presenting original data? I could not understand this from the text, as the authors do not explicitly devote space to this clarification.
Minor comments:
All Latin names should be italicised, but in this manuscript they almost never are, e.g. L11, L21, L208, L209, L212, and many other places.
Animal species should always be named with their Latin name (in addition to the common English one), e.g., L179 or in Table 3, where only informal names are used.
L57: Excluding the abstract, this is the first mention of BSFL, so the Latin name should appear here. In contrast, it is only introduced in full on L76, which suggests another failure in the final manuscript editing. Similarly, after introducing the acronym (BSFL), the authors should consistently use it throughout the rest of the manuscript. However, we see it reintroduced awkwardly on L176, as if the acronym were new. Similarly, the full name “black soldier fly larvae” is used again on L255 and L311, which gives the impression that parts of the text were generated separately and stitched together without proper harmonisation.
Likewise, “defatted BSFL” are mentioned earlier in the text, but only actually explained to the reader on L301.